# Separable Operator Networks

**Xinling Yu**[*]                                                                              *xyu644@ucsb.edu*
*University of California, Santa Barbara*

**Sean Hooten**[*]                                                                      *sean.hooten@hpe.com*
*Hewlett Packard Labs, Hewlett Packard Enterprise*

**Ziyue Liu**                                                                             *ziyueliu@ucsb.edu*
*University of California, Santa Barbara*

**Yequan Zhao**                                                                      *yequan_zhao@ucsb.edu*
*University of California, Santa Barbara*

**Marco Fiorentino**                                                              *marco.fiorentino@hpe.com*
*Hewlett Packard Labs, Hewlett Packard Enterprise*

**Thomas Van Vaerenbergh**                                          *thomas.van-vaerenbergh@hpe.com*
*Hewlett Packard Labs, HPE Belgium*

**Zheng Zhang**                                                                     *zhengzhang@ece.ucsb.edu*
*University of California, Santa Barbara*

[*]*Contributed equally*

**Reviewed on OpenReview:** *https://openreview.net/forum?id=RYtJmFDAxv*

## Abstract

Operator learning has become a powerful tool in machine learning for modeling complex physical systems governed by partial differential equations (PDEs). Although Deep Operator Networks (DeepONet) show promise, they require extensive data acquisition. Physics-informed DeepONets (PI-DeepONet) mitigate data scarcity but suffer from inefficient training processes. We introduce Separable Operator Networks (SepONet), a novel framework that significantly enhances the efficiency of physics-informed operator learning. SepONet uses independent trunk networks to learn basis functions separately for different coordinate axes, enabling faster and more memory-efficient training via forward-mode automatic differentiation. We provide a universal approximation theorem for SepONet proving the existence of a separable approximation to any nonlinear continuous operator. Then, we comprehensively benchmark its representational capacity and computational performance against PI-DeepONet. Our results demonstrate SepONet's superior performance across various nonlinear and inseparable PDEs, with SepONet's advantages increasing with problem complexity, dimension, and scale. For 1D time-dependent PDEs, SepONet achieves up to $112\times$ faster training and $82\times$ reduction in GPU memory usage compared to PI-DeepONet, while maintaining comparable accuracy. For the 2D time-dependent nonlinear diffusion equation, SepONet efficiently handles the complexity, achieving a $6.44\%$ mean relative $\ell_2$ test error, while PI-DeepONet fails due to memory constraints. This work paves the way for extreme-scale learning of continuous mappings between infinite-dimensional function spaces. Open source code is available at `https://github.com/HewlettPackard/separable-operator-networks`.

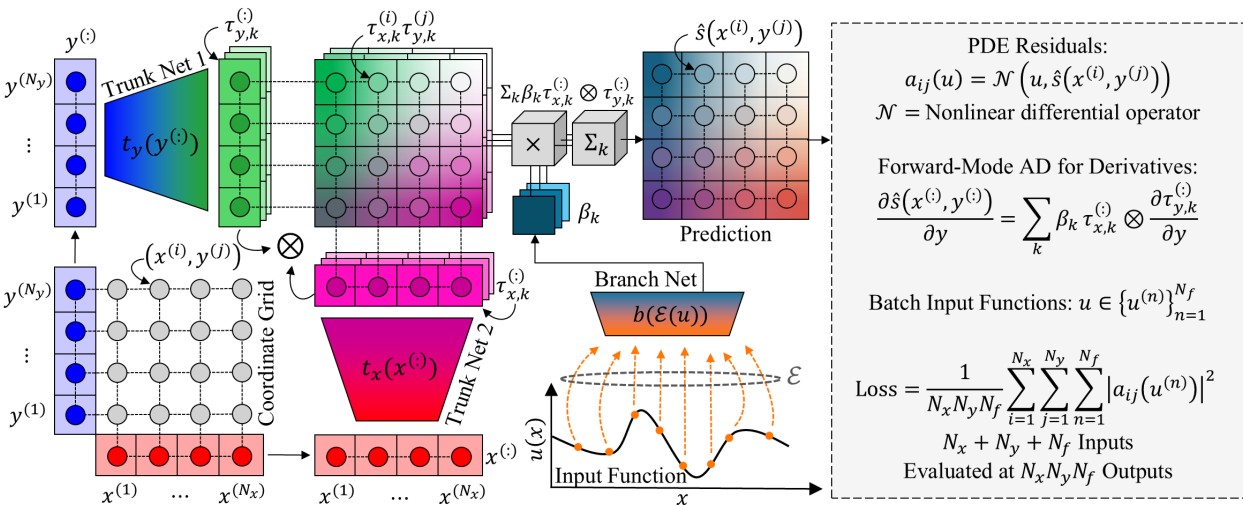

Figure 1: Separable operator network (SepONet) architecture for 2D problem instance. A coordinate grid of collocation points $(x^{(i)}, y^{(j)})$ can be evaluated efficiently by separating the coordinate axes, feeding them through independent trunk networks, and combining the outputs by outer product to obtain multiple basis function maps. Meanwhile, the branch network processes input functions and outputs coefficients, which are then used to scale and combine the trunk network basis functions by product and sum. Spatiotemporal derivatives of the output predictions are obtained efficiently by forward-mode automatic differentiation due to the independence of trunk networks along each coordinate axis.

# 1 Introduction

Operator learning, which aims to learn mappings between infinite-dimensional function spaces, has gained significant attention in scientific machine learning thanks to its ability to model complex dynamics in physics systems. This approach has been successfully applied to a wide range of applications, including climate modeling (Kashinath et al., 2021; Pathak et al., 2022), multiphysics simulation (Liu et al., 2023; Cai et al., 2021; Mao et al., 2021; Lin et al., 2021; Kontolati et al., 2024), inverse design (Lu et al., 2022b; Gu et al., 2022) and more (Shukla et al., 2023; Gupta & Brandstetter, 2022). Various operator learning algorithms (Lu et al., 2021; Li et al., 2020b;a; Ovadia et al., 2023; Wen et al., 2022; Ashiqur Rahman et al., 2022) have been developed to address these applications, with Deep Operator Networks (DeepONets) (Lu et al., 2021) being particularly notable due to their universal approximation guarantee for operators (Chen & Chen, 1995; Lanthaler et al., 2022; Gopalani et al.) and robustness (Lu et al., 2022a).

To approximate the function operator $G : \mathcal{U} \to \mathcal{S}$, DeepONets are trained in a supervised manner using a dataset of $N_f$ pair functions $\left\{u^{(i)}, s^{(i)}\right\}_{i=1}^{N_f}$, where in the context of parametric partial differential equations (PDEs), each $u^{(i)}$ represents a PDE configuration function and each $s^{(i)}$ represents the corresponding solution. Unlike traditional numerical methods which require repeated simulations for each different PDE configuration, once well trained, a DeepONet allows for efficient parallel inference in abstract infinite-dimensional function spaces. Given new PDE configurations, it can immediately provide the corresponding solutions. However, this advantage comes with a significant challenge: to achieve satisfactory generalization error, the number of required input-output function pairs grows quadratically (Lu et al., 2021; Lanthaler et al., 2022; Liu et al., 2022a; Gopalani et al.). Generating enough function pairs can be computationally expensive or even impractical in some applications, creating a bottleneck in the effective deployment of DeepONets.

Physics-informed deep operator networks (PI-DeepONet) (Wang et al., 2021b) have been introduced as a solution to the costly data acquisition problem. Inspired by physics-informed neural networks (PINNs) (Raissi et al., 2019), PI-DeepONet learns operators by constraining the DeepONet outputs to approximately satisfy the underlying governing PDE system parameterized by the input function $u$. This is achieved by penalizing the physics loss (including PDE residual loss, initial loss and boundary loss), thus eliminating

the need for ground-truth output functions $s$. However, PI-DeepONet shares the same disadvantage as PINNs in that the training process is both memory-intensive and time-consuming (He et al., 2023; Cho et al., 2024). This inefficiency, common to both PI-DeepONet and PINNs, arises from the need to compute high-order derivatives of the PDE predictions with respect to numerous collocation points during physics loss optimization. This computation typically relies on reverse-mode automatic differentiation (Baydin et al., 2018), involving the backpropagation of physics loss through the computational graph to update model parameters. The inefficiency is even more pronounced for PI-DeepONet, as it requires evaluating the physics loss across multiple PDE configurations. While numerous studies (He et al., 2023; Zhao et al., 2023; Hu et al., 2024; Liu et al., 2022b; Cho et al., 2024) have proposed methods to enhance PINN training efficiency, there has been limited research focused on improving the training efficiency of PI-DeepONet specifically.

To address the inefficiency in training PI-DeepONet, we propose Separable Operator Networks (SepONet), inspired by the separation of variables technique in solving PDEs and recent work on separable PINN (Cho et al., 2024). Suppose we want to approximate a nonlinear continuous operator of $d$ variables ($d$-dimensional collocation points). When optimizing the physics loss at $M$ collocation points, PI-DeepONet always requires $M$ inputs to evaluate PDE predictions and their derivatives, regardless of whether the points are on a regular grid or randomly sampled. This approach becomes inefficient when $M$ is large. By contrast, SepONet uses $d$ independent trunk networks to learn univariate basis functions separately for each variable, which can then be combined to obtain predictions in $d$ dimensions. In particular, if $M$ can be decomposed as $N_1 \times N_2 \times \cdots \times N_d$, where $N_n$ is the number of points sampled along the $n$-th coordinate axis (e.g., $1024 = 16 \times 8 \times 8$ for $d = 3$), SepONet requires only $N_1 + N_2 + \cdots + N_d$ inputs to evaluate all $M$ collocation points on a regular grid, resulting in a more efficient solution. The SepONet architecture for a $d = 2$ problem instance is shown in Figure 1. This simple yet effective modification enables fast training and memory-efficient implementation of SepONet by leveraging forward-mode automatic differentiation (Khan & Barton, 2015) to compute high-order derivatives of all $N_1 \times N_2 \times \cdots \times N_d$ collocation points. It's important to note that factorizing functions in high-dimensional domains into multiple sub-functions defined over one-dimensional domains has been explored in supervised operator learning methods (Tran et al., 2021; Li et al., 2024; Kossaifi et al., 2023). Working independently and concurrently, our work and that of Mandl et al. (2025) extended these ideas to physics-informed operator learning frameworks. While both approaches leverage separable structures to enhance computational efficiency, our method uniquely draws from classical separation of variables techniques in PDEs, leading to a distinct low-rank functional decomposition of the trunk network. Our key contributions are:

1. We introduce SepONet, a physics-informed operator learning framework that significantly enhances training time and GPU memory usage relative to PI-DeepONet, enabling extreme-scale learning of continuous mappings between infinite-dimensional function spaces.

2. We provide a theoretical foundation for SepONet through the universal approximation theorem, proving its capability to approximate any nonlinear continuous operator with arbitrary accuracy.

3. We provide extensive benchmarks validating SepONet's representational capacity and computational performance relative to PI-DeepONet on a range of 1D and 2D time-dependent nonlinear PDEs. Our findings reveal that scaling up SepONet in both number of functions and collocation points consistently improves its accuracy, while typically outperforming PI-DeepONet. On 1D time-dependent PDEs, we achieve up to $112\times$ training speed-up with minimal memory increase. Notably, at moderately large scales where training PI-DeepONet exhausts 80GB of GPU memory, SepONet trains and operates efficiently with less than 1GB. Furthermore, we observe efficient scaling of SepONet with problem dimension, enabling accurate prediction of 2D time-dependent PDEs at scales where PI-DeepONet fails.

## 2 Preliminaries

### 2.1 Operator Learning for Solving Parametric Partial Differential Equations

Let $\mathcal{X}$ and $\mathcal{Y}$ be Banach spaces, with $K \subseteq \mathcal{X}$ and $K_1 \subseteq \mathcal{Y}$ being compact sets. Consider a nonlinear continuous operator $G : \mathcal{U} \to \mathcal{S}$, mapping functions from one infinite-dimensional space to another, where

$\mathcal{U} \subseteq C(K)$ and $\mathcal{S} \subseteq C(K_1)$. The goal of operator learning is to approximate the operator $G$ using a model parameterized by $\boldsymbol{\theta}$, denoted as $G_{\boldsymbol{\theta}}$. Here, $\mathcal{U}$ and $\mathcal{S}$ represent spaces of functions where the input and output functions have dimensions $d_u$ and $d_s$, respectively. We focus on the scalar case where $d_u = d_s = 1$ throughout most of this paper; however, it should be noted that the results apply to arbitrary $d_u$ and $d_s$.

In the context of solving parametric partial differential equations (PDEs), consider PDEs of the form:

$$\mathcal{N}(u, s) = 0, \ \mathcal{I}(u, s) = 0, \ \mathcal{B}(u, s) = 0, \tag{1}$$

where $\mathcal{N}$ is a nonlinear differential operator, $\mathcal{I}$ and $\mathcal{B}$ represent the initial and boundary conditions, $u \in \mathcal{U}$ denotes the PDE configurations (source terms, coefficients, initial conditions, and etc.), and $s \in \mathcal{S}$ denotes the corresponding PDE solution. Assuming that for any $u \in \mathcal{U}$ there exists a unique solution $s \in \mathcal{S}$, we can define the solution operator $G : \mathcal{U} \to \mathcal{S}$ as $s = G(u)$.

A widely used framework for approximating such an operator $G$ involves constructing $G_{\boldsymbol{\theta}}$ through three maps (Lanthaler et al., 2022):

$$G_{\boldsymbol{\theta}} \approx G := \mathcal{D} \circ \mathcal{A} \circ \mathcal{E}. \tag{2}$$

First, the encoder $\mathcal{E} : \mathcal{U} \to \mathbb{R}^m$ maps an input function $u \in \mathcal{U}$ to a finite-dimensional feature representation. Next, the approximator $\mathcal{A} : \mathbb{R}^m \to \mathbb{R}^r$ transforms this encoded data within the finite-dimensional space $\mathbb{R}^m$ to another finite-dimensional space $\mathbb{R}^r$. Finally, the decoder $\mathcal{D} : \mathbb{R}^r \to \mathcal{S}$ produces the output function $s(y) = G(u)(y)$ for $y \in K_1$.

## 2.2 Deep Operator Networks (DeepONet)

The original DeepONet formulation (Lu et al., 2021) can be analyzed through the 3-step approximation framework (2). The encoder $\mathcal{E} : \mathcal{U} \to \mathbb{R}^m$ maps the input function $u$ to its point-wise evaluations at $m$ fixed sensors $x_1, x_2, \ldots, x_m \in K$, e.g., $(u(x_1), ..., u(x_m)) = \mathcal{E}(u)$. Two separate neural networks (usually multilayer perceptrons), the branch net and the trunk net, serve as the approximator and decoder, respectively. The branch net $b_{\psi} : \mathbb{R}^m \to \mathbb{R}^r$ parameterized by $\psi$ processes $(u(x_1), \ldots, u(x_m))$ to produce a feature embedding $(\beta_1, \beta_2, \ldots, \beta_r)$. The trunk net $t_{\phi} : \mathbb{R}^d \to \mathbb{R}^r$ with parameters $\phi$, takes a continuous coordinate $y = (y_1, ..., y_d)$ as input and outputs a feature embedding $(\tau_1, \tau_2, \ldots, \tau_r)$. The final DeepONet prediction of a function $u$ for a query $y$ is:

$$G_{\boldsymbol{\theta}}(u)(y) = \sum_{k=1}^{r} \beta_k \tau_k = b_{\psi}(\mathcal{E}(u)) \cdot t_{\phi}(y), \tag{3}$$

where $\cdot$ is the vector dot product and $\boldsymbol{\theta} = (\psi, \phi)$ represents all the trainable parameters in the branch and trunk nets.

Despite DeepONet's remarkable success across a range of applications in multiphysics simulation (Cai et al., 2021; Mao et al., 2021; Lin et al., 2021), inverse design (Lu et al., 2022b), and carbon storage (Jiang et al., 2023), its supervised training process is highly dependent on the availability of training data, which can be costly. Indeed, the generalization error of DeepONets scales quadratically with the number of training input-output function pairs (Lu et al., 2021; Lanthaler et al., 2022; Liu et al., 2022a; Gopalani et al.). Generating a large number of high-quality training data is expensive or even impractical in some applications. For example, in simulating high Reynolds number ($Re$) turbulent flow (Pope, 2001), accurate numerical simulations require a fine mesh, leading to a computational cost scaling with $Re^3$ (Kochkov et al., 2021), making the generation of sufficiently large and diverse training datasets prohibitively expensive.

To address the need for costly data acquisition, physics-informed deep operator networks (PI-DeepONet) (Wang et al., 2021b), inspired by physics-informed neural networks (PINNs) (Raissi et al., 2019), have been proposed to learn operators without relying on observed input-output function pairs. Given a dataset of $N_f$ input training functions, $N_r$ residual points, $N_I$ initial points, and $N_b$ boundary points: $\mathcal{D} = \left\{ \{u^{(i)}\}_{i=1}^{N_f}, \{y_r^{(j)}\}_{j=1}^{N_r}, \{y_I^{(j)}\}_{j=1}^{N_I}, \{y_b^{(j)}\}_{j=1}^{N_b} \right\}$, PI-DeepONets are trained by minimizing an unsupervised physics loss:

$$L_{physics}(\boldsymbol{\theta}|\mathcal{D}) = L_{residual}(\boldsymbol{\theta}|\mathcal{D}) + \lambda_I L_{initial}(\boldsymbol{\theta}|\mathcal{D}) + \lambda_b L_{boundary}(\boldsymbol{\theta}|\mathcal{D}), \tag{4}$$

where

$$L_{residual}(\boldsymbol{\theta}|\mathcal{D}) = \frac{1}{N_f N_r} \sum_{i=1}^{N_f} \sum_{j=1}^{N_r} \left| \mathcal{N}(u^{(i)}, G_{\boldsymbol{\theta}}(u^{(i)})(y_r^{(j)})) \right|^2,$$

$$L_{initial}(\boldsymbol{\theta}|\mathcal{D}) = \frac{1}{N_f N_I} \sum_{i=1}^{N_f} \sum_{j=1}^{N_I} \left| \mathcal{I}(u^{(i)}, G_{\boldsymbol{\theta}}(u^{(i)})(y_I^{(j)})) \right|^2, \tag{5}$$

$$L_{boundary}(\boldsymbol{\theta}|\mathcal{D}) = \frac{1}{N_f N_b} \sum_{i=1}^{N_f} \sum_{j=1}^{N_b} \left| \mathcal{B}(u^{(i)}, G_{\boldsymbol{\theta}}(u^{(i)})(y_b^{(j)})) \right|^2.$$

Here, $\lambda_I$ and $\lambda_b$ denote the weight coefficients for different loss terms. However, as noted in the original PI-DeepONet paper (Wang et al., 2021b), the training process can be both memory-intensive and time-consuming. Similar to PINNs (Raissi et al., 2019), this inefficiency arises because optimizing the physics loss requires calculating high-order derivatives of the PDE solution with respect to numerous collocation points, typically achieved via reverse-mode automatic differentiation (Baydin et al., 2018). This process involves backpropagating the physics loss through the unrolled computational graph to update the model parameters. For PI-DeepONet, the inefficiency is even more pronounced, as the physics loss terms (equation (5)) must be evaluated across multiple PDE configurations. Although various works (Chiu et al., 2022; He et al., 2023; Cho et al., 2024) have proposed different methods to improve the training efficiency of PINNs, little research has focused on enhancing the training efficiency of PI-DeepONet. We propose to address this inefficiency through a separation of input variables.

### 2.3 Separation of Variables

The method of separation of variables seeks solutions to PDEs of the form $s(y) = T(t)Y_1(y_1)\cdots Y_d(y_d)$ for an input point $y = (t, y_1, \ldots, y_d)$ and univariate functions $T, Y_1, \ldots, Y_d$. Suppose we have a linear PDE system

$$\mathcal{M}[t]s(y) = \mathcal{L}_1[y_1]s(y) + \cdots + \mathcal{L}_d[y_d]s(y), \tag{6}$$

where $\mathcal{M}[t] = \frac{d}{dt} + h(t)$ is a first order differential operator of $t$, and $\mathcal{L}_1[y_1],...,\mathcal{L}_d[y_d]$ are linear second order ordinary differential operators of their respective variables $y_1,...,y_d$ only. Furthermore, assume we are provided Robin boundary conditions in each variable and separable initial condition $s(t = 0, y_1, \ldots, y_d) = \prod_{n=1}^{d} \phi_n(y_n)$ for functions $\phi_n(y_n)$ that satisfy the boundary conditions. Then, leveraging Sturm-Liouville theory and some massaging, the solution to this problem can be written

$$s(y) = s(t, y_1, \ldots, y_d) = \sum_k A_k T^k(t) \prod_{n=1}^{d} Y_n^k(y_n), \tag{7}$$

where $k$ is a lumped index that counts over infinite eigenfunctions of each $\mathcal{L}_i$ operator (potentially with repeats). For example, given $n \in \{1,...,d\}$, $\mathcal{L}_n Y_n^k(y_n) = \lambda_n^k Y_n^k$ for eigenvalue $\lambda_n^k \in \mathbb{R}$. $T^k(t)$ depends on all the eigenvalues $\lambda_n^k$ corresponding to index $k$. $A_k \in \mathbb{R}$ is a coefficient determined by the initial condition. More details can be found in Appendix C. The method of separation of variables applied to a linear heat equation example can be found in Appendix D.2.

One may notice the resemblance between the form of the DeepONet prediction in (3) with (7), provided $\beta_k = A_k$ and $\tau_k = T^k(t) \prod_{i=1}^{d} Y_i^k(y_i)$, with appropriately ordered $k$. We leverage this similarity explicitly in the construction of SepONet below.

## 3 Separable Operator Networks (SepONet)

Inspired by the method of separation of variables (7) and recent work on separable PINN (Cho et al., 2024), we propose using separable operator networks (SepONet) to learn basis functions separately for different coordinate axes. SepONet approximates the solution operator of a PDE system by, for given point

$y = (y_1, \ldots, y_d),$

$$
\begin{aligned}
G_{\boldsymbol{\theta}}(u)(y_1, \ldots, y_d) &= \sum_{k=1}^{r} \beta_k \prod_{n=1}^{d} \tau_{n,k} \\
&= b_{\psi}(\mathcal{E}(u)) \cdot \left( t_{\phi_1}^1(y_1) \odot t_{\phi_2}^2(y_2) \odot \cdots \odot t_{\phi_d}^d(y_d) \right),
\end{aligned}
\tag{8}
$$

where $\odot$ is the Hadamard (element-wise) vector product and $\cdot$ is the vector dot product. Here, $\beta_k = b_{\psi}(\mathcal{E}(u))_k$ is the $k$-th output of the branch net, as in DeepONet. However, unlike DeepONet, which employs a single trunk net that processes each collocation point $y$ individually, SepONet uses $d$ independent trunk nets, $t_{\phi_n}^n : \mathbb{R} \to \mathbb{R}^r$ for $n = 1, \ldots, d$. In particular, $\tau_{n,k} = t_{\phi_n}^n(y_n)_k$ denotes the $k$-th output of the $n$-th trunk net. Importantly, the parameters of the $n$-th trunk net $\phi_n$ are independent of all other trunk net parameter sets. Viewed through the 3-step approximation framework (2), SepONet and DeepONet have identical encoder and approximator but different decoders.

Equation (8) can be understood as a low-rank approximation of the solution operator by truncating the basis function series (represented by the output shape of the trunk nets) at a maximal number of ranks $r$. SepONet not only enjoys the advantage that basis functions for different variables with potentially different scales can be learned more easily and efficiently, but also allows for fast and efficient training by leveraging forward-mode automatic differentiation, which we will discuss in Section 3.1 and Section 3.2. Moreover, despite the resemblance between (8) and the separation of variables method for linear PDEs (7) (discussed below in Section 3.1.3), we find that SepONet can effectively approximate the solutions to nonlinear parametric PDEs. Indeed, we provide a universal approximation theorem for separable operator networks in Section 3.3 and extensive accuracy and performance scaling numerical experiments for nonlinear and inseparable PDEs in Section 4.

Finally, it is worth noting that if one is only interested in solving deterministic PDEs under a certain configuration (i.e., $u$ is fixed), then the coefficients $\beta_k$ are also fixed and can be absorbed by the basis functions. In this case, SepONet will reduce to separable PINN Cho et al. (2024), which has been proven to be efficient and accurate in solving single-scenario PDE systems (Es' kin et al., 2024; Oh et al., 2024).

## 3.1 SepONet Architecture and Implementation Details

Suppose we are provided a computation domain $K_1 = [0, 1]^d$ of dimension $d$ and an input function $u$. To evaluate the residual loss term in equation (5) on $N_r$ random collocation points, PI-DeepONet samples all $N_r$ points directly from $K_1$. However, if $N_r$ can be approximately factorized as $N_1 \times N_2 \times \cdots \times N_d$ (e.g., $1024 = 16 \times 8 \times 8$ for $d = 3$), and we relax the Monte Carlo sampling requirement for $d$-dimensional collocation points, SepONet only needs to randomly sample $N_n$ points along the $n$-th coordinate axis, resulting in a total of $N_1 + N_2 + \cdots + N_d$ samples.

It is important to note that SepONet's mapping from $N_1 + N_2 + \cdots + N_d$ inputs to $N_1 \times N_2 \times \cdots \times N_d$ outputs is most efficient when using regular grid sampling. However, we have empirically demonstrated (in Section 4) that this per-axis grid-based sampling strategy does not degrade SepONet's accuracy compared to PI-DeepONet's Monte Carlo random sampling over the entire domain $K_1$. Irregular domains may be sampled by (a) dividing the irregular domain into subdomains each approximated by a regular grid, or (b) applying a coordinate transformation to map the irregular domain onto a regular one (e.g., converting Cartesian to polar coordinates for a circular domain).

For shorthand and generality, we will denote the dataset of input points for SepONet as $\mathcal{D} = \{y_1^{(:)}, \ldots, y_d^{(:)}\}$. Each $y_n^{(:)} = \{y_n^{(i)}\}_{i=1}^{N_n}$ represents an array of $N_n$ samples along the $n$-th coordinate axis for a total of $N_1 + N_2 + \cdots + N_d$ samples. The initial and boundary points may be separately sampled from $\partial K_1$; the number of samples ($N_I$ and $N_b$) and sampling strategy are equivalent for SepONet and PI-DeepONet.

### 3.1.1 Forward Pass

The forward pass of SepONet, illustrated for $d = 2$ in Figure 1, follows the formulation (8) except generalized to the computationally advantageous setting where predictions along a grid of collocation points are processed

in parallel. The formula can be expressed:

$$
\begin{aligned}
G_{\boldsymbol{\theta}}(u)(y_1^{(:)}, \ldots, y_d^{(:)}) &= \sum_{k=1}^{r} \beta_k \bigotimes_{n=1}^{d} \tau_{n,k}^{(:)} \\
&= \sum_{k=1}^{r} b_{\psi}(\mathcal{E}(u))_k \left( t_{\phi_1}^1(y_1^{(:)})_k \otimes t_{\phi_2}^2(y_2^{(:)})_k \otimes \cdots \otimes t_{\phi_d}^d(y_d^{(:)})_k \right),
\end{aligned}
\tag{9}
$$

where $\otimes$ is the (outer) tensor product, which produces an output predictive array along a meshgrid of $N_1 \times N_2 \times \cdots \times N_d$ collocation points. Notably, $\tau_{n,k}^{(:)} = t_{\phi_n}^n(y_n^{(:)})_k$ represents a vector of $N_n$ values produced by the $n$-th trunk net along the $k$-th output mode after feeding all $y_n^{(:)}$ points. After taking the outer product along each of $n = 1, \ldots, d$ dimensions for all $r$ modes, the modes are sum-reduced with the predictions of the branch net $\beta_k = b_{\psi}(\mathcal{E}(u))_k$. While not shown here, our implementation also batches over input functions $\{u^{(i)}\}_{i=1}^{N_f}$ for $N_f$ functions. Thus, for only $N_f + N_1 + \cdots + N_d$ inputs, SepONet produces a predictive array with shape $N_f \times N_1 \times \cdots \times N_d$.

### 3.1.2 Model Update

In evaluation of the physics loss (4), SepONet enables more efficient computation of high-order derivatives in terms of both time and memory use compared to PI-DeepONet by leveraging forward-mode automatic differentiation (AD) (Khan & Barton, 2015). This is fairly evident by the form of (9). For example, to compute derivatives of all SepONet outputs with respect to the $m$-th variable $y_m$:

$$
\frac{\partial G_{\boldsymbol{\theta}}(u)(y_1^{(:)}, ..., y_m^{(:)}, ..., y_d^{(:)})}{\partial y_m} = \sum_{k=1}^{r} \beta_k \left( \bigotimes_{n \neq m} \tau_{n,k}^{(:)} \right) \otimes \frac{\partial \tau_{m,k}^{(:)}}{\partial y_m},
\tag{10}
$$

where $\frac{\partial \tau_{m,k}^{(:)}}{\partial y_m}$ is a vector of derivatives of the $m$-th trunk net's $k$-th basis function evaluated along all inputs to the $m$-th coordinate axis. One may notice that $\frac{\partial \tau_{m,k}^{(:)}}{\partial y_m}$ can be written as a Jacobian-vector product (JVP) of the Jacobian of the $m$-th trunk net's $r \times N_m$ outputs with respect to all $N_m$ inputs, and a length $N_m \times 1$ tangent vector of 1's:

$$
\frac{\partial \tau_{m,k}^{(:)}}{\partial y_m} := e^{(k)} \mathbf{J}[t_{\phi_m}^m(y_m^{(:)})]\mathbf{1},
\tag{11}
$$

where $e^{(k)}$ selects the $k$-th output mode from the resulting $r \times N_m$ JVP output. This is equivalent to forward-mode AD. Consequently, the derivatives along the $m$-th coordinate axis across the entire grid of predictions can be obtained by pushing forward derivatives of the $m$-th trunk net, and then reusing the outputs of all other $n \neq m$ trunk nets via outer product. By contrast, PI-DeepOnet must compute derivatives $\partial G_{\boldsymbol{\theta}}(y_1^{(i)}, ..., y_d^{(i)})/\partial y_m$ for each input-output pair individually, $y^{(i)} = (y_1^{(i)}, ..., y_d^{(i)})$ for $i = 1, ..., M$, where there is no such computational advantage and it is more prudent to use reverse-mode AD. Fundamentally, the advantage of SepONet for using forward-mode AD can be attributed to the significantly smaller input-output relationship when evaluating along coordinate grids $\mathbb{R}^{N_1 + \cdots + N_d} \to \mathbb{R}^{N_1 \times \cdots \times N_d}$ compared to PI-DeepONet $\mathbb{R}^{M \times d} \to \mathbb{R}^{M \times 1}$ when we choose $M = N_1 N_2 \cdots N_d$. The time and space complexity analysis below in Section 3.2 provides a more descriptive breakdown of computational scaling behavior. For a more detailed explanation of forward- and reverse-mode AD, we refer readers to Cho et al. (2024); Margossian (2019). Once the physics loss is computed, often involving multiple evaluations of (10), reverse-mode AD is employed to update the model parameters $\boldsymbol{\theta} = (\psi, \phi_1, \ldots, \phi_d)$.

### 3.1.3 Inference

Once trained, SepONet can efficiently map input functions $u$ to accurate output function predictions $\hat{s} = G_{\boldsymbol{\theta}}(u)$ along large spatiotemporal grids. This is achieved by combining the learned coefficients from the

Table 1: Time and space complexities of first-order derivatives of SepONet and PI-DeepONet with respect to $N^d$ collocation points using forward-mode and reverse-mode AD, respectively.

| Method | Time Complexity | Space Complexity |
|---|---|---|
| SepONet (Forward AD) | $O(N \cdot d \cdot Lr^2 + N^d \cdot rd)$ | $O(N \cdot rd + N^d)$ |
| PI-DeepONet (Reverse AD) | $O(N^d \cdot Lr^2)$ | $O(N^d \cdot Lr)$ |

branch net with the basis functions learned by the trunk nets. Indeed, intriguing comparisons can be made between the results of the method of separation of variables (7) and trained SepONet predictions (8). For an initial value problem with linear, separable PDE operator treated in Section 2.3, we might expect SepONet to learn initial value function dependent coefficients $b_\psi(\mathcal{E}(u))_k = A_k$ and spatiotemporal basis functions $t_{\phi_n}^n(y_n)_k = Y_n^k(y_n)$ (provided we supply one additional trunk net $t_{\phi_0}^0(t)_k = T^k(t)$ for the temporal dimension) for appropriately ordered modes $k$ and sufficiently large $r$. Examples of the learned basis functions for a separable 1D time-dependent heat equation initial value problem example are provided in Appendix D.2 as a function of the number of the trunk net output shape $r$. For a small number of modes $r = 1$ or $r = 2$, SepONet learns nearly the exact spatiotemporal basis functions obtained by separation of variables. For larger $r$, the SepONet basis functions do not converge to the analytically expected basis functions. Nevertheless, approximation error is observed to improve with $r$, and near perfect accuracy is obtained at large $r$ by comparison to numerical estimates of the analytic solution.

While the form of SepONet predictions (8) resemble separation of variables, we note that the method of separation of variables typically only applies to linear PDEs with restricted properties. In spite of this, SepONet is capable of accurately approximating arbitrary operator learning problems (including nonlinear PDEs) as guaranteed by a universal approximation property, provided below in Section 3.3.

## 3.2 Complexity Analysis

Suppose we are provided a computational domain $K_1 = [0, 1]^d$ of dimension $d$. For PI-DeepONet, collocation points are sampled randomly from the entire $d$-dimensional domain, with a total of $M$ points. For SepONet, as described previously in Section 3.1, we randomly sample $N_1 + N_2 + \cdots + N_d$ inputs for $N_n$ points along the $n$-th axis, and construct a Cartesian product grid in $K_1$ via (9). The resulting output of PI-DeepONet has shape $M \times 1$, and the output of SepONet has shape $N_1 \times N_2 \times \cdots \times N_d$. For simplicity, we assume all trunk nets are $L$-layer fully connected networks with hidden and output dimensions $r$, and that $N_1 = N_2 = \cdots = N_d = N$, and $M = N^d$. The resulting time and space complexity to compute first-order derivatives of all SepONet and PI-DeepONet outputs is provided in Table 1.

The first term in SepONet's time and space complexity is due to the forward-mode AD computation of each of the $d$ trunk networks derivatives with respect to $N$ inputs per axis. The second term, containing $N^d$, is from computing and storing the tensor product. On the other hand, PI-DeepONet individually backpropagates all $M = N^d$ outputs, resulting in complexity scaling with $N^d$.

From this analysis, in the limiting case of $N^{d-1} \gg Lr$, we observe that both SepONet and PI-DeepONet have time and space complexities that include $N^d$ due to evaluations over all points in a $d$-dimensional space. However, since typically $d \ll Lr$, SepONet is more efficient in practice due to smaller coefficients in the $N^d$ term. Moreover, the tensor product in SepONet can be greatly accelerated by GPU parallelization, which is not taken into account in this analysis. In the limiting case $Lr \gg N^{d-1}$, the first term in SepONet's time complexity dominates $O(N \cdot d \cdot Lr^2)$, or in other words, it scales linearly with dimension and sub-linearly with the total number of collocation points $N^d$. This situation is not uncommon in many practical 2D and 3D operator learning problems.

Note that Table 1 only considered first-order derivatives. Higher-order derivatives are typically needed to evaluate physics loss functions (5). Fortunately, higher-order derivatives for SepONet are computed with similar complexity to Table 1, since they amount to sequentially repeating the Jacobian-vector products (JVP) from (10) and (11). Lastly, we did not consider the parameter update for training with a physics loss

in Table 1, since it requires further assumptions about the composition of the branch network, and it is not typically the limiting computation for either PI-DeepONet or SepONet.

### 3.3 Universal Approximation Property of SepONet

The universal approximation property of DeepONet has been discussed in Chen & Chen (1995); Lu et al. (2021). Here we present the universal approximation theorem to show that proposed separable operator networks can also approximate any nonlinear continuous operators that map infinite-dimensional function spaces to others.

**Theorem 1** (Universal Approximation Theorem for Separable Operator Networks). *Suppose that $\sigma$ is a Tauber-Wiener function, $g$ is a sinusoidal function, $\mathcal{X}$ is a Banach space, $K \subseteq \mathcal{X}$, $K_1 \subseteq \mathbb{R}^{d_1}$ and $K_2 \subseteq \mathbb{R}^{d_2}$ are three compact sets in $\mathcal{X}$, $\mathbb{R}^{d_1}$ and $\mathbb{R}^{d_2}$, respectively, $\mathcal{U}$ is a compact set in $C(K)$, $G$ is a nonlinear continuous operator, which maps $\mathcal{U}$ into a compact set $\mathcal{S} \subseteq C(K_1 \times K_2)$, then for any $\epsilon > 0$, there are positive integers $n$, $r$, $m$, constants $c_i^k$, $\zeta_k^1$, $\zeta_k^2$, $\xi_{ij}^k$, $\theta_i^k \in \mathbb{R}$, points $\omega_k^1 \in \mathbb{R}^{d_1}$, $\omega_k^2 \in \mathbb{R}^{d_2}$, $x_j \in K$, $i = 1, \ldots, n$, $k = 1, \ldots, r$, $j = 1, \ldots, m$, such that*

$$\left| G(u)(y) - \sum_{k=1}^{r} \underbrace{\sum_{i=1}^{n} c_i^k \sigma \left( \sum_{j=1}^{m} \xi_{ij}^k u(x_j) + \theta_i^k \right)}_{branch} \underbrace{g\left( w_k^1 \cdot y_1 + \zeta_k^1 \right)}_{trunk_1} \underbrace{g\left( w_k^2 \cdot y_2 + \zeta_k^2 \right)}_{trunk_2} \right| < \epsilon \tag{12}$$

*holds for all $u \in \mathcal{U}$, $y = (y_1, y_2) \in K_1 \times K_2$.*

*Proof.* The proof can be found in Appendix A.2. $\qquad\square$

**Remark 1.** *The definition of the Tauber-Wiener function is given in Appendix A.1. It is worth noting that many common-used activations, such as ReLU, GELU and Tanh, are Tauber-Wiener functions.*

**Remark 2.** *Here we show the approximation property of a separable operator network with two trunk nets, by repeatedly applying trigonometric angle addition formula, it is trivial to separate $y$ as $(y_1, y_2, \ldots, y_d) \in K_1 \times K_2 \times \ldots \times K_d$ and extend (12) to $d$ trunk nets.*

**Remark 3.** *In our assumptions, we restrict the activation function for the trunk nets to be sinusoidal. This choice is motivated by the natural suitability of sinusoidal functions for constructing basis functions (Stein & Shakarchi, 2011) and their empirical effectiveness in solving PDEs (Sitzmann et al., 2020). However, it would be interesting to explore whether Theorem 1 still holds when $g$ is a more general activation function, such as a Tauber-Wiener function. We will leave this investigation for future work.*

**Remark 4.** *Here we assume a two-layer branch network and $d$ one-layer trunk networks with sinusoidal activations. For practical implementation, our theoretical results can be extended to multi-layer trunk networks by leveraging the Universal Approximation Theorem (UAT) to approximate sinusoidal functions.*

**Remark 5.** *Theorem 1 suggests the existence of a separable operator network approximation for any nonlinear continuous operator. This does not imply error bounds nor tractable scaling laws with respect to any specific error metric, nor does it provide a prescription for how to define and update model parameters. Below we will provide experimental evidence that error scaling is comparable to PI-DeepONet when using physics-informed operator learning. Please note that error bounds for the supervised training of DeepONet have been previously derived by Lanthaler et al. (2022); similar error bounds for SepONet are not provided in this work.*

## 4 Numerical Results

This section presents comprehensive numerical studies demonstrating the expressive power and effectiveness of SepONet compared to PI-DeepONet on various time-dependent PDEs: diffusion-reaction, advection, Burgers', and (2+1)-dimensional nonlinear diffusion equations. Both models were trained by optimizing the

Table 2: Summary of PDE test problems.

| Governing Law | Domain | Equation Form | Initial Condition | Boundary Condition |
|---|---|---|---|---|
| Diffusion-reaction | $x \in [0,1]$, $t \in [0,1]$ | $\frac{\partial s}{\partial t} = D \frac{\partial^2 s}{\partial x^2} + k s^2 + u(x)$ | $s(x,0) = 0$ | $s(0,t) = 0$, $s(1,t) = 0$ |
| Advection | $x \in [0,1]$, $t \in [0,1]$ | $\frac{\partial s}{\partial t} + u(x) \frac{\partial s}{\partial x} = 0$ | $s(x,0) = \sin(\pi x)$ | $s(0,t) = \sin\left(\frac{\pi}{2} t\right)$ |
| Burgers' | $x \in [0,1]$, $t \in [0,1]$ | $\frac{\partial s}{\partial t} + s \frac{\partial s}{\partial x} - \nu \frac{\partial^2 s}{\partial x^2} = 0$ | $s(x,0) = u(x)$ | $s(0,t) = s(1,t)$, $\frac{\partial s}{\partial x}(0,t) = \frac{\partial s}{\partial x}(1,t)$ |
| 2D Nonlinear diffusion | $\boldsymbol{x} \in \Omega = [0,1]^2$, $t \in [0,1]$ | $\frac{\partial s}{\partial t} = \alpha \nabla \cdot (s \nabla s)$ | $s(\boldsymbol{x},0) = u(\boldsymbol{x})$ | $s(\boldsymbol{x},t) = 0$ on $\partial \Omega$ |

physics loss (equation (4)) on a dataset $\mathcal{D}$ consisting of input functions, residual points, initial points, and boundary points. PDE definitions are summarized in Table 2.

We set the number of residual points to $N_r = N^d = N_c$, where $d$ is the problem dimension and $N$ is an integer. Here, $N_c$ refers to the total number of training points. For SepONet, the residual points are generated by randomly sampling $N$ points along each axis and constructing a Cartesian product grid. In contrast, for PI-DeepONet, the $N^d$ residual points are randomly sampled from the entire $d$-dimensional domain. The number of initial and boundary points per axis is set to $N_I = N_b = N = \sqrt[d]{N_c}$, and these points are also randomly sampled from the solution domain. For each PDE, the model size remains fixed as $N_c$ or $N_f$ varies. Specifically, both PI-DeepONet and SepONet have branch and trunk networks of the same size; the main difference is that SepONet uses $d$ independent trunk networks, one for each axis.

We evaluate both models by varying the number of input functions ($N_f$) and training points ($N_c$) across four key perspectives: test accuracy, GPU memory usage, training time, and extreme-scale learning capabilities. The main results are illustrated in Figure 2 and Figure 3, with complete test results reported in Appendix B.3. Loss functions, training details, and problem-specific parameters are provided in Appendix B.1 and Appendix B.2. We provide additional ablation studies for our test results using trunk networks with hyperbolic tangent activation functions in Appendix B.4.1, and varied number of input sensors for the branch net in Appendix B.4.2.

## 4.1 Test accuracy

Both PI-DeepONet and SepONet demonstrate improved accuracy when increasing either the number of training points ($N_c$) or the number of input functions ($N_f$), while fixing the other parameter. This trend is consistent across all four equations tested.

For instance, in the case of the diffusion-reaction equation, when fixing $N_f = 100$ and varying $N_c$ from $8^2$ to $128^2$, the relative $\ell_2$ error of PI-DeepONet decreases from 1.39% to 0.73%, while SepONet's error reduces from 1.49% to 0.62%. Conversely, when fixing $N_c = 128^2$ and varying $N_f$ from 5 to 100, PI-DeepONet's error drops from 34.54% to 0.73%, and SepONet's reduces from 22.40% to 0.62%.

## 4.2 GPU memory usage

While both models show improved accuracy with increasing $N_c$ or $N_f$, their memory usage patterns differ significantly. This divergence is particularly evident in the case of the advection equation.

When fixing $N_f = 100$ and varying $N_c$, PI-DeepONet exhibits a steep increase in GPU memory consumption during training, rising from 0.967 GB at $N_c = 8^2$ to 59.806 GB at $N_c = 128^2$. In contrast, SepONet maintains a relatively constant and low memory footprint during training, ranging between 0.713 GB and 0.719 GB across the same range of $N_c$. Similarly, when fixing $N_c = 128^2$ and varying $N_f$ from 5 to 100, PI-DeepONet's memory usage escalates from 3.021 GB to 59.806 GB. SepONet, however, maintains a stable memory usage throughout this range.

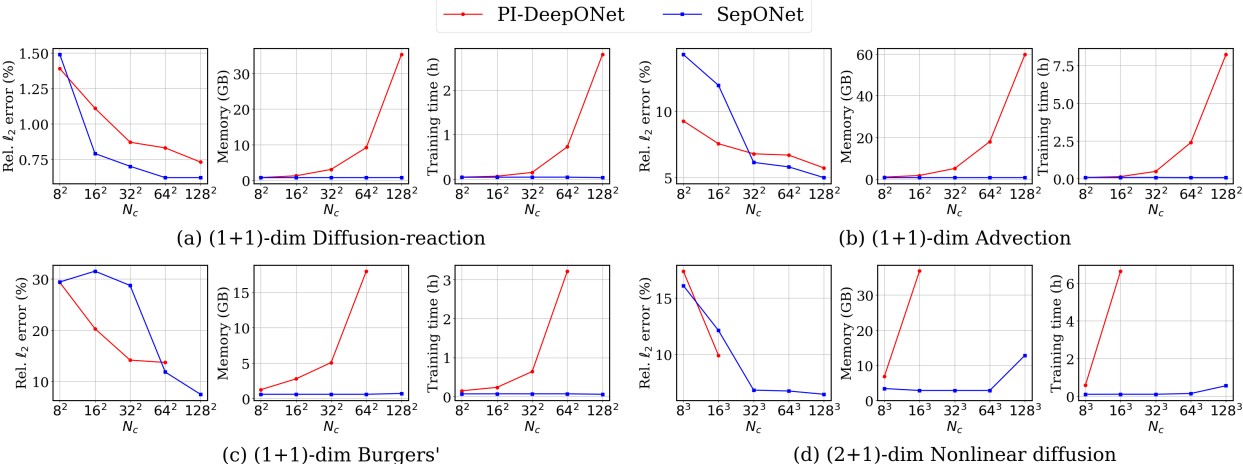

Figure 2: Performance comparison of PI-DeepONet and SepONet with varying number of training points ($N_c$) and fixed number of input functions ($N_f = 100$). Results show test accuracy, GPU memory usage, and training time for four PDEs. As $N_c$ increases, both models demonstrate improved accuracy, but PI-DeepONet exhibits significant increases in training time and memory usage, while SepONet maintains better computational efficiency.

### 4.3 Training time

The training time scaling exhibits a pattern similar to memory usage, as demonstrated by the advection equation example. As $N_c$ increases from $8^2$ to $128^2$ with fixed $N_f = 100$, PI-DeepONet's training time increases significantly from 0.0787 to 8.231 hours (2.361 to 246.93 ms per iteration). In contrast, SepONet maintains relatively stable training times, ranging from 0.0730 to 0.0843 hours (2.19 to 2.529 ms per iteration) over the same $N_c$ range. Similarly, when varying $N_f$ from 5 to 100 with fixed $N_c = 128^2$, PI-DeepONet's training time increases from 0.3997 to 8.231 hours (11.991 to 246.93 ms per iteration). SepONet, however, keeps training times between 0.0730 and 0.0754 hours. These results demonstrate SepONet's superior scalability in terms of training time. The ability to maintain near-constant training times across a wide range of problem sizes is a significant advantage, particularly for large-scale applications where computational efficiency is crucial.

### 4.4 Extreme-scale learning

The Burgers' and nonlinear diffusion equations highlight SepONet's capabilities in extreme-scale learning scenarios.

For the Burgers' equation, PI-DeepONet encounters memory limitations at larger scales. As seen in Figure 2(c), PI-DeepONet can only compute up to $N_c = 64^2$, achieving a relative $\ell_2$ error of 13.72%. In contrast, SepONet continues to improve, reaching a 7.51% error at $N_c = 128^2$. The nonlinear diffusion equation further emphasizes this difference. In Figure 3(d), PI-DeepONet results are entirely unavailable due to out-of-memory issues. SepONet, however, efficiently handles this complex problem, achieving a relative $\ell_2$ error of 6.44% with $N_c = 128^3$ and $N_f = 100$. Table 3 demonstrates SepONet's ability to tackle even larger scales for the Burgers' equation. It achieves a relative $\ell_2$ error as low as 4.12% with $N_c = 512^2$ and $N_f = 800$, while maintaining reasonable memory usage (10.485 GB) and training time (0.478 hours). These results underscore SepONet's capability to handle extreme-scale learning problems beyond the reach of PI-DeepONet due to computational constraints.

## 5 Discussion

The field of operator learning faces a critical dilemma. Deep Operator Networks (DeepONets) offer fast training but require extensive data generation, which can be prohibitively expensive or impractical for

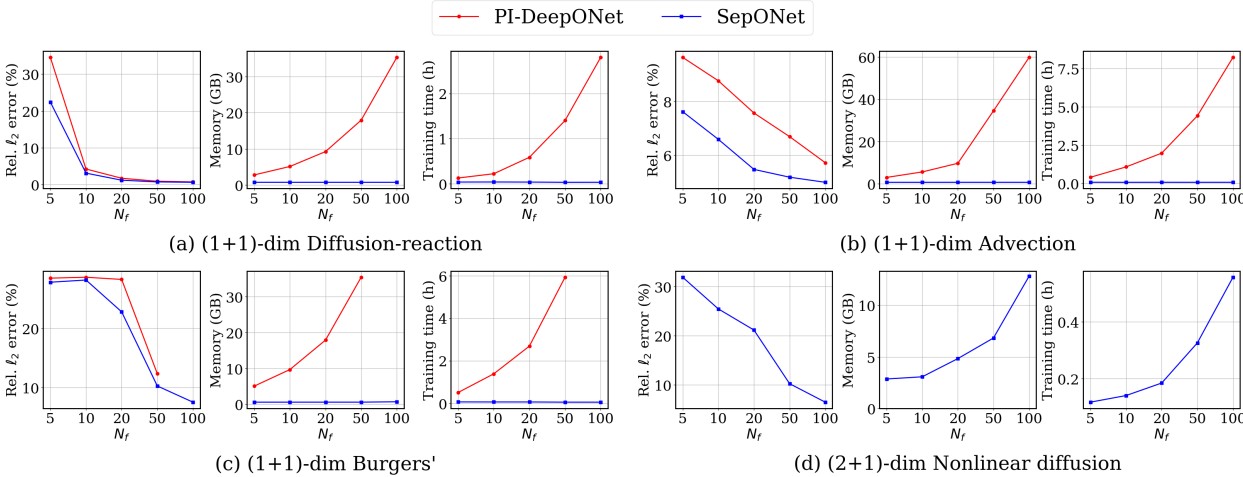

Figure 3: Performance comparison of PI-DeepONet and SepONet with increasing number of input functions ($N_f$) and fixed number of training points ($N_c = 128^d$, where $d$ is the problem dimension). Both models show improved accuracy with increasing $N_f$, but PI-DeepONet's computational resources scale poorly compared to SepONet's more efficient scaling. Note: PI-DeepONet results for the (2+1)-dimensional diffusion equation are unavailable due to memory constraints.

Table 3: Additional SepONet results for Burgers' equation, demonstrating that larger $N_c$ and $N_f$ can be used to enhance accuracy with minimal cost increase.

| Metrics \ $N_c$ & $N_f$ | $128^2$ & 400 | $128^2$ & 800 | $256^2$ & 400 | $256^2$ & 800 | $512^2$ & 400 | $512^2$ & 800 |
|---|---|---|---|---|---|---|
| Relative $\ell_2$ error (%) | 6.60 | 6.21 | 5.68 | 4.46 | 5.38 | 4.12 |
| Memory (GB) | 0.966 | 1.466 | 2.466 | 4.466 | 5.593 | 10.485 |
| Training time (hours) | 0.0771 | 0.0957 | 0.1238 | 0.1717 | 0.2751 | 0.478 |

complex systems. Physics-informed DeepONets (PI-DeepONets) relax the data requirement but at the cost of resource-intensive training processes. This creates a challenging trade-off: balancing resource allocation either before training (data generation) or during training (computational resources).

Our proposed Separable Operator Networks (SepONet) effectively address both of these concerns. Inspired by the separation of variables technique typically used for linear PDEs, SepONet constructs its own basis functions to approximate a nonlinear operator. SepONet's expressive power is guaranteed by the universal approximation property (Theorem 1), ensuring it can approximate any nonlinear continuous operator with arbitrary accuracy. Our numerical results corroborate this theoretical guarantee, demonstrating SepONet's ability to handle complex, nonlinear systems efficiently.

By leveraging independent trunk networks for different variables, SepONet enables an efficient implementation via forward-mode automatic differentiation (AD). This approach achieves remarkable efficiency in both data utilization and computational resources. SepONet is trained solely by optimizing the physics loss, eliminating the need for expensive simulations to generate ground truth PDE solutions. In terms of computational resources, SepONet maintains stable GPU memory usage and training time, even with increasing training data and network size, in contrast to PI-DeepONet's dramatic resource consumption increases under similar scaling. We anticipate that SepONet's advantages will allow it to tackle more challenging physics-informed operator learning problems, such as the Navier-Stokes equations (Jin et al., 2021), where both input and output functions are vector-valued. These are problems that PI-DeepONet may struggle to train on due to resource constraints. As an example, we have considered a (2+1)-dimensional Navier-Stokes equation, previously investigated in the context of PINN (Cho et al., 2024; Wang et al., 2024). Some early, preliminary results can be found in Appendix D.1.

However, SepONet has certain limitations that warrant further research and development. The mesh grid structure of SepONet's solution, while enabling efficient training through forward-mode AD, may limit its flexibility in handling PDEs with irregular geometries. Addressing this limitation could involve developing adaptations or hybrid approaches that accommodate more complex spatial domains (Li et al., 2023; Serrano et al., 2024; Fang et al., 2024), potentially expanding SepONet's applicability to a broader range of physical problems.

Additionally, while the linear decoder allows for an efficient SepONet implementation, a very large number of basis functions may be needed for accurate linear representation in some problems (Seidman et al., 2022). Developing a nonlinear decoder version of SepONet will be useful to balance accuracy and efficient training. Moreover, implementing an adaptive weighting strategy (Wang et al., 2021a; 2022a;b) for different loss terms in the physics loss function, instead of using predefined fixed weights, could lead to improved accuracy and faster convergence.

Finally, empirical observations suggest that training accuracy and robustness improve with an increase in input training functions. However, the neural scaling laws in physics-informed operator learning remain unexplored, presenting an intriguing theoretical challenge for future investigation.

## Acknowledgments

We would like to thank Wolfger Peelaers for the valuable discussions and insightful comments. Z. Zhang was supported by NSF Awards #2328281 and #1846476

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

# A  Universal Approximation Theorem for Separable Operator Networks

Here we present the universal approximation theorem for the proposed separable operator networks, originally written in Theorem 1 and repeated below in Theorem 5. We begin by reviewing established theoretical results on approximating continuous functions and functionals. Following this review, we introduce the preliminary lemmas and proofs necessary for understanding Theorem 5. We refer our readers to Chen & Chen (1995); Weierstrass (1885) for detailed proofs of Theorems 2, 3, 4. Main notations are listed in Table 4.

## A.1  Preliminaries and Auxiliary Results

**Definition 1** (Tauber-Wiener (TW)). *If a function $g : \mathbb{R} \to \mathbb{R}$ (continuous or discontinuous) satisfies that all the linear combinations $\sum_{i=1}^{N} c_i g\left(\lambda_i x + \theta_i\right)$, $\lambda_i \in \mathbb{R}$, $\theta_i \in \mathbb{R}$, $c_i \in \mathbb{R}$, $i = 1, 2, \ldots, N$, are dense in every $C[a,b]$, then $g$ is called a Tauber-Wiener (TW) function.*

**Remark 1** (Density in $C[a,b]$). *A set of functions is said to be dense in $C[a,b]$ if every function in the space of continuous functions on the interval $[a,b]$ can be approximated arbitrarily closely by functions from the set.*

**Definition 2** (Compact Set). *Suppose that $X$ is a Banach space, $V \subseteq X$ is called a compact set in $X$, if for every sequence $\{x_n\}_{n=1}^{\infty}$ with all $x_n \in V$, there is a subsequence $\{x_{n_k}\}$, which converges to some element $x \in V$.*

**Theorem 2** (Chen & Chen (1995)). *Suppose that $K$ is a compact set in $\mathbb{R}^n$, $\mathcal{S}$ is a compact set in $C(K)$, $g \in (TW)$, then for any $\epsilon > 0$, there exist a positive integer $N$, real numbers $\theta_i$, vectors $\omega_i \in \mathbb{R}^n$, $i = 1, \ldots, N$, which are independent of $f \in C(K)$ and constants $c_i(f)$, $i = 1, \ldots, N$ depending on $f$, such that*

$$\left| f(x) - \sum_{i=1}^{N} c_i(f) g\left(\omega_i \cdot x + \theta_i\right) \right| < \epsilon \tag{13}$$

*holds for all $x \in K$ and $f \in \mathcal{S}$. Moreover, each $c_i(f)$ is a linear continuous functional defined on $\mathcal{S}$.*

**Theorem 3** (Chen & Chen (1995)). *Suppose that $\sigma \in (TW)$, $X$ is a Banach Space, $K \subseteq X$ is a compact set, $\mathcal{U}$ is a compact set in $C(K)$, $f$ is a continuous functional defined on $\mathcal{U}$, then for any $\epsilon > 0$, there are positive integers $N$, $m$ points $x_1, \ldots, x_m \in K$, and real constants $c_i$, $\theta_i$, $\xi_{ij}$, $i = 1, \ldots, N$, $j = 1, \ldots, m$, such that*

$$\left| f(u) - \sum_{i=1}^{N} c_i \sigma\left(\sum_{j=1}^{m} \xi_{ij} u\left(x_j\right) + \theta_i\right) \right| < \epsilon \tag{14}$$

*holds for all $u \in \mathcal{U}$.*

**Theorem 4** (Weierstrass Approximation Theorem Weierstrass (1885)). *Suppose $f \in C[a,b]$, then for every $\epsilon > 0$, there exists a polynomial $p$ such that for all $x$ in $[a,b]$, we have $|f(x) - p(x)| < \epsilon$.*

**Corollary 1.** *Trigonometric polynomials are dense in the space of continuous and periodic functions $\tilde{C}[0, 2\pi] := \{f \in C[0, 2\pi] \mid f(0) = f(2\pi)\}$.*

*Proof.* For any $\tilde{f} \in \tilde{C}[0, 2\pi]$, extend it to a $2\pi$-periodic and continuous function $f$ defined on $\mathbb{R}$. It suffices to show that there exists a trigonometric polynomial that approximates $f$ within any $\epsilon > 0$. We construct the continuous even functions of $2\pi$ period $g$ and $h$ as:

$$g(\theta) = \frac{f(\theta) + f(-\theta)}{2} \quad \text{and} \quad h(\theta) = \frac{f(\theta) - f(-\theta)}{2} \sin(\theta). \tag{15}$$

Let $\phi(x) = g(\arccos x)$ and $\psi(x) = h(\arccos x)$. Since $\phi, \psi$ are continuous functions on $[-1, 1]$, by the Weierstrass Approximation Theorem 4, for any $\epsilon > 0$, there exist polynomials $p$ and $q$ such that

$$|\phi(x) - p(x)| < \frac{\epsilon}{4} \quad \text{and} \quad |\psi(x) - q(x)| < \frac{\epsilon}{4} \tag{16}$$

Table 4: Notations and Symbols

| | |
|---|---|
| $X$ | some Banach space with norm $\|\cdot\|_X$ |
| $\mathbb{R}^d$ | Euclidean space of dimension $d$ |
| $K$ | some compact set in a Banach space |
| $C(K)$ | Banach space of all continuous functions defined on $K$, with norm $\|f\|_{C(K)} = \max_{x\in K}|f(x)|$ |
| $\tilde{C}[a,b]$ | the space of functions in $C[a,b]$ satisfying $f(a) = f(b)$ |
| $V$ | some compact set in $C(K)$ |
| $u(x)$ | some input function |
| $\mathcal{U}$ | the space of input functions |
| $G$ | some continuous operator |
| $G(u)(y)$ or $s(y)$ | some output function that is mapped from the corresponding input function $u$ by the operator $G$ |
| $\mathcal{S}$ | the space of output functions |
| (TW) | all the Tauber-Wiener functions |
| $\sigma$ and $g$ | activation function for branch net and trunk nets in Theorem 5 |
| $\{x_1, x_2, \ldots, x_m\}$ | $m$ sensor points for identifying input function $u$ |
| $r$ | rank of some deep operator network or separable operator network |
| $n, m$ | operator network size hyperparameters in Theorem 5 |

holds for all $x \in [-1, 1]$. Let $x = \cos\theta$, it follows that

$$|g(\theta) - p(\cos\theta)| < \frac{\epsilon}{4} \quad \text{and} \quad |h(\theta) - q(\cos\theta)| < \frac{\epsilon}{4} \tag{17}$$

for $\theta \in [0, \pi]$. Because $g$, $h$ and *cosine* are even and $2\pi$-periodic, (17) holds for all $\theta \in \mathbb{R}$. From the definitions of $g$ and $h$, and the fact $|\sin\theta| \le 1$, $|\sin^2\theta| \le 1$, we have

$$\left|\frac{f(\theta) + f(-\theta)}{2}\sin^2\theta - p(\cos\theta)\sin^2\theta\right| < \frac{\epsilon}{4} \quad \text{and} \quad \left|\frac{f(\theta) - f(-\theta)}{2}\sin^2\theta - q(\cos\theta)\sin\theta\right| < \frac{\epsilon}{4}. \tag{18}$$

Using the triangle inequality, we obtain

$$\left|f(\theta)\sin^2\theta - \left[p(\cos\theta)\sin^2\theta + q(\cos\theta)\sin\theta\right]\right| < \frac{\epsilon}{2}. \tag{19}$$

Applying the same analysis to

$$\tilde{g}(\theta) = \frac{f(\theta + \frac{\pi}{2}) + f(-\theta + \frac{\pi}{2})}{2} \quad \text{and} \quad \tilde{h}(\theta) = \frac{f(\theta + \frac{\pi}{2}) - f(-\theta + \frac{\pi}{2})}{2}\sin(\theta), \tag{20}$$

we can find polynomials $r$ and $s$ such that

$$\left|f\left(\theta + \frac{\pi}{2}\right)\sin^2\theta - \left[r(\cos\theta)\sin^2\theta + s(\cos\theta)\sin\theta\right]\right| < \frac{\epsilon}{2} \tag{21}$$

holds for all $\theta$. Substituting $\theta$ with $\theta - \frac{\pi}{2}$ gives

$$\left|f(\theta)\cos^2\theta - \left[r(\sin\theta)\cos^2\theta - s(\sin\theta)\cos\theta\right]\right| < \frac{\epsilon}{2}. \tag{22}$$

By the triangle inequality, combining (22) and (19) gives

$$\left|f(\theta) - \left[r(\sin\theta)\cos^2\theta - s(\sin\theta)\cos\theta + p(\cos\theta)\sin^2\theta + q(\cos\theta)\sin\theta\right]\right| < \epsilon \tag{23}$$

holds for all $\theta$. Thus, the trigonometric polynomial

$$r(\sin\theta)\cos^2\theta - s(\sin\theta)\cos\theta + p(\cos\theta)\sin^2\theta + q(\cos\theta)\sin\theta \tag{24}$$

is an $\epsilon$-approximation to $f$. $\qquad\square$

**Remark 2.** *If $p(x)$ is a polynomial, it is easy to verify that $p(\cos\theta)$ is a trigonometric polynomial due to the fact $\cos^n\theta = \sum_{k=0}^{n} \frac{\binom{n}{k}}{2^n}\cos\left((n-2k)\theta\right)$.*

Prior to proving Theorem 5, we need to establish the following lemmas.

**Lemma 1.** *Sine is a Tauber-Wiener function.*

*Proof.* Assuming the interval to be $[0,\pi]$ first. For every continuous function $f$ on $[0,\pi]$ and any $\epsilon > 0$, we can extend $f$ to a continuous function $F$ on $[0,2\pi]$ so that $F(x) = f(x)$ on $[0,\pi]$ and $F(2\pi) = F(0)$. By Lemma 1, there exists a trigonometric polynomial

$$p(x) = a_0 + \sum_{n=1}^{N} a_n \cos(nx) + b_n \sin(nx) \tag{25}$$

such that

$$\sup_{x \in [0,\pi]} |f(x) - p(x)| \leq \sup_{x \in [0,2\pi]} |F(x) - p(x)| < \epsilon. \tag{26}$$

Let $c_0 = a_0$, $\lambda_0 = 0$, $\theta_0 = \frac{\pi}{2}$, $c_{2n-1} = b_n$, $\lambda_{2n-1} = n$, $\theta_{2n-1} = 0$, $c_{2n} = a_n$, $\lambda_{2n} = n$, $\theta_{2n} = \frac{\pi}{2}$, for $n = 1, 2, \ldots, N$, $p(x)$ is redefined as

$$p(x) = \sum_{i=0}^{2N} c_i \sin\left(\lambda_i x + \theta_i\right). \tag{27}$$

Thus we have

$$\left| f(x) - \sum_{i=0}^{2N} c_i \sin\left(\lambda_i x + \theta_i\right) \right| < \epsilon \tag{28}$$

for $x \in [0,\pi]$. Now consider a continuous function $g$ on $[a,b]$, define $f(x) \in C[0,\pi] := g\left(\frac{b-a}{\pi}x + a\right)$, then by (28), we have

$$\left| g(x) - \sum_{i=0}^{2N} c_i \sin\left(\frac{\pi\lambda_i}{b-a}x - \frac{\pi\lambda_i a}{b-a} + \theta_i\right) \right| < \epsilon \tag{29}$$

holds for all $x \in [a,b]$. Therefore, it follows that for any continuous function $g$ on $[a,b]$ and any $\epsilon > 0$, we can approximate $g$ within $\epsilon$ by choosing $N$ sufficiently large and adjusting $c_i$, $\lambda_i$, $\theta_i$ accordingly. Hence, the set of all such linear combinations of $\sin(x)$ is dense in $C[a,b]$, confirming that $\sin(x)$ is a Tauber-Wiener function. $\qquad\square$

**Remark 3.** *It is straightforward to conclude that all sinusoidal functions are Tauber-Wiener functions.*

**Lemma 2.** *Suppose that $V_1 \subseteq X_1$, $V_2 \subseteq X_2$ are two compact sets in Banach spaces $X_1$ and $X_2$, respectively, then their Cartesian product $V_1 \times V_2$ is also compact.*

*Proof.* For every sequence $\{x_n^1, x_n^2\}$ in $V_1 \times V_2$, since $V_1$ is compact, $\{x_n^1\}$ has a subsequence $\{x_{n_k}^1\}$ that converges to some element $x^1 \in V_1$. As well, since $V_2$ is compact, there exists a subsequence $\{x_{n_k}^2\}$ that converges to $x^2 \in V_2$. It follows that $\{x_n^1, x_n^2\}$ converges to $\left(x^1, x^2\right) \in V_1 \times V_2$, thus $V_1 \times V_2$ is compact. $\quad\square$

**Lemma 3.** *Suppose that $X$ is a Banach space, $K_1 \subseteq X_1$, $K_2 \subseteq X_2$ are two compact sets in $X_1$ and $X_2$, respectively. $\mathcal{U}$ is a compact set in $C(K_1)$, then the range $G(\mathcal{U})$ of the continuous operator $G$ from $\mathcal{U}$ to $C(K_2)$ is compact in $C(K_2)$.*

*Proof.* For every sequence $\{f_n\}$ in $\mathcal{U}$, since $\mathcal{U}$ is compact, there exists a subsequence $\{f_{n_k}\}$ that converges to some function $f \in \mathcal{U}$. Since $G$ is continuous, the convergence $f_{n_k} \to f$ in $C(K_1)$ implies

$$G(f_{n_k}) \to G(f) \quad \text{in} \quad C(K_2). \tag{30}$$

Thus, for every sequence $\{G(f_n)\}$ in $G(\mathcal{U})$, there exists a subsequence $\{G(f_{n_k})\}$ that converges to $G(f) \in G(\mathcal{U})$. Therefore, the range $G(\mathcal{U})$ of the continuous operator $G$ is compact in $C(K_2)$. $\qquad\square$

## A.2 Universal Approximation Theorem for SepONet

**Theorem 5** (Universal Approximation Theorem for Separable Operator Networks). *Suppose that $\sigma \in$ (TW), $g$ is a sinusoidal function, $X$ is a Banach Space, $K \subseteq X$, $K_1 \subseteq \mathbb{R}^{d_1}$ and $K_2 \subseteq \mathbb{R}^{d_2}$ are three compact sets in $X$, $\mathbb{R}^{d_1}$ and $\mathbb{R}^{d_2}$, respectively, $\mathcal{U}$ is a compact set in $C(K)$, $G$ is a nonlinear continuous operator, which maps $\mathcal{U}$ into a compact set $\mathcal{S} \subseteq C(K_1 \times K_2)$, then for any $\epsilon > 0$, there are positive integers $n$, $r$, $m$, constants $c_i^k$, $\zeta_k^1$, $\zeta_k^2$, $\xi_{ij}^k$, $\theta_i^k \in \mathbb{R}$, points $\omega_k^1 \in \mathbb{R}^{d_1}$, $\omega_k^2 \in \mathbb{R}^{d_2}$, $x_j \in K_1$, $i = 1, \ldots, n$, $k = 1, \ldots, r$, $j = 1, \ldots, m$, such that*

$$\left| G(u)(y) - \sum_{k=1}^{r} \sum_{i=1}^{n} c_i^k \sigma \left( \sum_{j=1}^{m} \xi_{ij}^k u(x_j) + \theta_i^k \right) g\left(w_k^1 \cdot y_1 + \zeta_k^1\right) g\left(w_k^2 \cdot y_2 + \zeta_k^2\right) \right| < \epsilon \tag{31}$$

*holds for all $u \in \mathcal{U}$, $y = (y_1, y_2) \in K_1 \times K_2$.*

*Proof.* Without loss of generality, we can assume that $g$ is sine function, by Lemma 1, we have $g \in (TW)$; From the assumption that $K_1$ and $K_2$ are compact, by Lemma 2, $K_1 \times K_2$ is compact; Since $G$ is a continuous operator that maps $\mathcal{U}$ into $C(K_1 \times K_2)$, it follows that the range $G(\mathcal{U}) = \{G(u) : u \in \mathcal{U}\}$ is compact in $C(K_1 \times K_2)$ due to Lemma 3; Thus by Theorem 2, for any $\epsilon > 0$, there exists a positive integer $N$, real numbers $c_k(G(u))$ and $\zeta_k$, vectors $\omega_k \in R^{d_1+d_2}$, $k = 1, \ldots, N$, such that

$$\left| G(u)(y) - \sum_{k=1}^{N} c_k(G(u)) g(\omega_k \cdot y + \zeta_k) \right| < \frac{\epsilon}{2} \tag{32}$$

holds for all $y \in K_1 \times K_2$ and $u \in C(K)$. Let $(\omega_k^1, \omega_k^2) = \omega_k$, where $\omega_k^1 \in \mathbb{R}^{d_1}$ and $\omega_k^2 \in \mathbb{R}^{d_2}$. Utilizing the trigonometric angle addition formula, we have

$$g(\omega_k \cdot y + \zeta_k) = g\left(\omega_k^1 \cdot y_1 + \zeta_k\right) g\left(\omega_k^2 \cdot y_2 + \frac{\pi}{2}\right) + g\left(\omega_k^1 \cdot y_1 + \zeta_k + \frac{\pi}{2}\right) g\left(\omega_k^2 \cdot y_2\right). \tag{33}$$

Let $r = 2N$, $c_{N+k}(G(u)) = c_k(G(u))$, $\omega_{N+k}^1 = \omega_k^1$, $\omega_{N+k}^2 = \omega_k^2$, $\zeta_k^1 = \zeta_k$, $\zeta_k^2 = \frac{\pi}{2}$ for $k = 1, \ldots, N$, and let $\zeta_k^1 = \zeta_k + \frac{\pi}{2}$, $\zeta_k^2 = 0$ for $k = N + 1, \ldots, r$, equation 32 can be expressed as:

$$\left| G(u)(y) - \sum_{k=1}^{r} c_k(G(u)) g\left(\omega_k^1 \cdot y_1 + \zeta_k^1\right) g\left(\omega_k^2 \cdot y_2 + \zeta_k^2\right) \right| < \frac{\epsilon}{2}. \tag{34}$$

Since $G$ is a continuous operator, according to the last proposition of Theorem 2, we conclude that for each $k = 1, \ldots, 2N$, $c_k(G(u))$ is a continuous functional defined on $\mathcal{U}$. Repeatedly applying Theorem 3, for each $k = 1, \ldots, 2N$, $c_k(G(u))$, we can find positive integers $n_k, m_k$, constants $c_i^k$, $\xi_{ij}^k$, $\theta_i^k \in R$ and $x_j \in K_1$, $i = 1, \ldots, n_k$, $j = 1, \ldots, m_k$, such that

$$\left| c_k(G(u)) - \sum_{i=1}^{n_k} c_i^k \sigma \left( \sum_{j=1}^{m_k} \xi_{ij}^k u(x_j) + \theta_i^k \right) \right| < \frac{\epsilon}{2L} \tag{35}$$

holds for all $k = 1, \ldots, r$ and $u \in \mathcal{U}$, where

$$L = \sum_{k=1}^{r} \sup_{y_1 \in K_2, y_2 \in K_3} \left| g\left(\omega_k^1 \cdot y_1 + \zeta_k^1\right) g\left(\omega_k^2 \cdot y_2 + \zeta_k^2\right) \right|. \tag{36}$$

Substituting (35) into (34), we obtain that

$$\left| G(u)(y) - \sum_{k=1}^{r} \sum_{i=1}^{n_k} c_i^k \sigma \left( \sum_{j=1}^{m_k} \xi_{ij}^k u(x_j) + \theta_i^k \right) g\left(w_k^1 \cdot y_1 + \zeta_k^1\right) g\left(w_k^2 \cdot y_2 + \zeta_k^2\right) \right| < \epsilon \tag{37}$$

holds for all $u \in \mathcal{U}$, $y_1 \in K_1$ and $y_2 \in K_2$. Let $n = \max_k n_k$, $m = \max_k m_k$. For all $n_k < i \le n$, let $c_i^k = 0$. For all $m_k < j \le m$, let $\xi_{ij}^k = 0$. Then (37) can be rewritten as:

$$\left| G(u)(y) - \sum_{k=1}^{r} \sum_{i=1}^{n} c_i^k \sigma \left( \sum_{j=1}^{m} \xi_{ij}^k u\left(x_j\right) + \theta_i^k \right) g\left(w_k^1 \cdot y_1 + \zeta_k^1\right) g\left(w_k^2 \cdot y_2 + \zeta_k^2\right) \right| < \epsilon, \tag{38}$$

which holds for all $u \in \mathcal{U}$, $y_1 \in K_1$ and $y_2 \in K_2$. This completes the proof of Theorem 5.

$\square$

# B  PDE Problem Definitions, Training details, and Complete Test Results

## B.1  PDE Problem Definitions

All PDE test problems exhibited in Section 4 are described in the subsections below.

### B.1.1  Diffusion-Reaction Systems

We set the diffusion coefficient $D = 0.01$ and the reaction rate $k = 0.01$. The input training source terms are sampled from a mean-zero Gaussian random field (GRF) (Seeger, 2004) with a length scale 0.2. To generate the test dataset, we sample 100 different source terms from the same GRF and apply a second-order implicit finite difference method (Iserles, 2009) to obtain the reference solutions on a uniform $128 \times 128$ grid. The specific physics loss terms in equation (5) are defined as follows:

$$
\begin{aligned}
\mathcal{L}_{residual} &= \frac{1}{N_f N_r} \sum_{i=1}^{N_f} \sum_{j=1}^{N_r} \left| \frac{\partial G_{\boldsymbol{\theta}}(u^{(i)})(x_r^{(j)}, t_r^{(j)})}{\partial t_r^{(j)}} - D \frac{\partial^2 G_{\boldsymbol{\theta}}(u^{(i)})(x_r^{(j)}, t_r^{(j)})}{\partial (x_r^{(j)})^2} \right. \\
&\qquad\qquad \left. - k \left( G_{\boldsymbol{\theta}}(u^{(i)})(x_r^{(j)}, t_r^{(j)}) \right)^2 - u^{(i)}(x_r^{(j)}) \right|^2, \\
\mathcal{L}_{initial} &= \frac{1}{N_f N_I} \sum_{i=1}^{N_f} \sum_{j=1}^{N_I} \left| G_{\boldsymbol{\theta}}(u^{(i)})(x_I^{(j)}, 0) \right|^2, \\
\mathcal{L}_{boundary} &= \frac{1}{N_f N_b} \sum_{i=1}^{N_f} \sum_{j=1}^{N_b} \left( \left| G_{\boldsymbol{\theta}}(u^{(i)})(0, t_b^{(j)}) \right|^2 + \left| G_{\boldsymbol{\theta}}(u^{(i)})(1, t_b^{(j)}) \right|^2 \right).
\end{aligned}
\tag{39}
$$

### B.1.2  Advection Equation

The input training variable coefficients are strictly positive by defining $u(x) = v(x) - \min_x v(x) + 1$, where $v$ is sampled from a GRF with length scale 0.2. To create the test dataset, we generate 100 new coefficients in the same manner that are not used in training and apply the Lax–Wendroff scheme (Iserles, 2009) to solve the advection equation on a uniform $128 \times 128$ grid. The specific physics loss terms in equation (5) are defined as follows:

$$
\begin{aligned}
\mathcal{L}_{residual} &= \frac{1}{N_f N_r} \sum_{i=1}^{N_f} \sum_{j=1}^{N_r} \left| \frac{\partial G_{\boldsymbol{\theta}}(u^{(i)})(x_r^{(j)}, t_r^{(j)})}{\partial t_r^{(j)}} + u^{(i)}(x_r^{(j)}) \frac{\partial G_{\boldsymbol{\theta}}(u^{(i)})(x_r^{(j)}, t_r^{(j)})}{\partial x_r^{(j)}} \right|^2, \\
\mathcal{L}_{initial} &= \frac{1}{N_f N_I} \sum_{i=1}^{N_f} \sum_{j=1}^{N_I} \left| G_{\boldsymbol{\theta}}(u^{(i)})(x_I^{(j)}, 0) - \sin(\pi x_I^{(j)}) \right|^2, \\
\mathcal{L}_{boundary} &= \frac{1}{N_f N_b} \sum_{i=1}^{N_f} \sum_{j=1}^{N_b} \left| G_{\boldsymbol{\theta}}(u^{(i)})(0, t_b^{(j)}) - \sin\left(\frac{\pi}{2} t_b^{(j)}\right) \right|^2.
\end{aligned}
\tag{40}
$$

### B.1.3 Burgers' Equation

The input training initial conditions are sampled from a GRF $\sim \mathcal{N}\left(0, 25^2\left(-\Delta + 5^2 I\right)^{-4}\right)$ using the Chebfun package (Driscoll et al., 2014), satisfying the periodic boundary conditions. Synthetic test dataset consists of 100 unseen initial functions and their corresponding solutions, which are generated from the same GRF and are solved by spectral method on a $101 \times 101$ uniform grid using the spinOp library (Palani, 2024), respectively. The corresponding physics loss terms in equation (5) are defined as:

$$
\begin{aligned}
\mathcal{L}_{residual} &= \frac{1}{N_f N_r} \sum_{i=1}^{N_f} \sum_{j=1}^{N_r} \left| \frac{\partial G_{\boldsymbol{\theta}}(u^{(i)})(x_r^{(j)}, t_r^{(j)})}{\partial t_r^{(j)}} + G_{\boldsymbol{\theta}}(u^{(i)})(x_r^{(j)}, t_r^{(j)}) \frac{\partial G_{\boldsymbol{\theta}}(u^{(i)})(x_r^{(j)}, t_r^{(j)})}{\partial x_r^{(j)}} \right. \\
&\qquad \left. - \nu \frac{\partial^2 G_{\boldsymbol{\theta}}(u^{(i)})(x_r^{(j)}, t_r^{(j)})}{\partial (x_r^{(j)})^2} \right|^2, \\
\mathcal{L}_{initial} &= \frac{1}{N_f N_I} \sum_{i=1}^{N_f} \sum_{j=1}^{N_I} \left| G_{\boldsymbol{\theta}}(u^{(i)})(x_I^{(j)}, 0) - u^{(j)}(x_I^{(j)}) \right|^2, \\
\mathcal{L}_{boundary} &= \frac{1}{N_f N_b} \sum_{i=1}^{N_f} \sum_{j=1}^{N_b} \left( \left| G_{\boldsymbol{\theta}}(u^{(i)})(0, t_b^{(j)}) - G_{\boldsymbol{\theta}}(u^{(i)})(1, t_b^{(j)}) \right|^2 + \left| \frac{\partial G_{\boldsymbol{\theta}}(u^{(i)})(x, t_b^{(j)})}{\partial x}\bigg|_{x=0} - \frac{\partial G_{\boldsymbol{\theta}}(u^{(i)})(x, t_b^{(j)})}{\partial x}\bigg|_{x=1} \right|^2 \right).
\end{aligned}
\tag{41}
$$

### B.1.4 2D Nonlinear Diffusion Equation

The input training initial conditions are generated as a sum of Gaussian functions, parameterized as:

$$
u(x, y) = \sum_{i=1}^{3} A_i \exp[-w_i\{(x - x_i)^2 + (y - y_i)^2\}],
\tag{42}
$$

where $A_i \sim \mathcal{U}(0.2, 0.5)$ are amplitudes, $w_i \sim \mathcal{U}(10, 20)$ are width parameters, and $(x_i, y_i) \sim \mathcal{U}(-0.5, 0.5)^2$ are center coordinates. We also generate 100 unseen test initial conditions using this method. The nonlinear diffusion equation is then solved using explicit Adams method to obtain reference solutions on a uniform $101 \times 101$ spatial grid with 101 time points. Physics loss terms in equation (5) for this problem are:

$$
\begin{aligned}
\mathcal{L}_{residual} &= \frac{1}{N_f N_r} \sum_{i=1}^{N_f} \sum_{j=1}^{N_r} \left| \frac{\partial G_{\boldsymbol{\theta}}(u^{(i)})(\boldsymbol{x}_r^{(j)}, t_r^{(j)})}{\partial t_r^{(j)}} - \alpha \nabla \cdot \left( G_{\boldsymbol{\theta}}(u^{(i)})(\boldsymbol{x}_r^{(j)}, t_r^{(j)}) \nabla G_{\boldsymbol{\theta}}(u^{(i)})(\boldsymbol{x}_r^{(j)}, t_r^{(j)}) \right) \right|^2, \\
\mathcal{L}_{initial} &= \frac{1}{N_f N_I} \sum_{i=1}^{N_f} \sum_{j=1}^{N_I} \left| G_{\boldsymbol{\theta}}(u^{(i)})(\boldsymbol{x}_I^{(j)}, 0) - u^{(i)}(\boldsymbol{x}_I^{(j)}) \right|^2, \\
\mathcal{L}_{boundary} &= \frac{1}{N_f N_b} \sum_{i=1}^{N_f} \sum_{j=1}^{N_b} \left| G_{\boldsymbol{\theta}}(u^{(i)})(\boldsymbol{x}_b^{(j)}, t_b^{(j)}) \right|^2.
\end{aligned}
\tag{43}
$$

### B.2 Training Details and Hyperparameters

Both PI-DeepONet and SepONet were trained by minimizing the physics loss (equation(4)) using gradient descent with the Adam optimizer (Kingma & Ba, 2014). The initial learning rate is $1 \times 10^{-3}$ and decays by a factor of 0.9 every 1,000 iterations. Additionally, we resample input training functions and training points (including residual, initial, and boundary points) every 100 iterations.

Across all benchmarks and on both models (SepONet and PI-DeepONet), we apply Tanh activation for the branch net and Sine activation for the trunk net. We note that no extensive hyperparameter tuning was performed for either PI-DeepONet or SepONet. The code in this study is implemented using JAX and Equinox libraries (Bradbury et al., 2018; Kidger & Garcia, 2021), and all training was performed on a single NVIDIA A100 GPU with 80 GB of memory. Training hyperparameters are provided in Table 5.

Table 5: Training hyperparameters for different PDE benchmarks

| Hyperparameters \ PDEs | Diffusion-reaction | Advection | Burgers' | 2D Nonlinear diffusion |
|---|---|---|---|---|
| # of sensors | 128 | 128 | 101 | 10201 ($101 \times 101$) |
| Network depth | 5 | 6 | 7 | 7 |
| Network width | 50 | 100 | 100 | 128 |
| # of training iterations | 50k | 120k | 80k | 80k |
| Weight coefficients ($\lambda_I$ / $\lambda_b$) | 1 / 1 | 100 / 100 | 20 / 1 | 20 / 1 |

## B.3 Complete Test Results

We report the relative $\ell_2$ error, root mean squared error (RMSE), GPU memory usage and total training time as metrics to assess the performance of PI-DeepONet and SepONet. Specifically, the mean and standard deviation of the relative $\ell_2$ error and RMSE are calculated over all functions in the test dataset. The complete test results are shown in Table 6 and Table 7.

Table 6: Performance comparison of PI-DeepONet and SepONet with varying number of training points ($N_c$) and fixed number of input training functions ($N_f = 100$). The '-' symbol indicates that results are not available due to out-of-memory issues.

| Equations | Metrics | Models | $8^d$ | $16^d$ | $32^d$ | $64^d$ | $128^d$ |
|---|---|---|---|---|---|---|---|
| Diffusion-Reaction $d=2$ | Relative $\ell_2$ error (%) | PI-DeepONet | $1.39 \pm 0.71$ | $1.11 \pm 0.59$ | $0.87 \pm 0.41$ | $0.83 \pm 0.35$ | $0.73 \pm 0.34$ |
| | | SepONet | $1.49 \pm 0.82$ | $0.79 \pm 0.35$ | $0.70 \pm 0.33$ | $0.62 \pm 0.28$ | $0.62 \pm 0.26$ |
| | RMSE ($\times 10^{-2}$) | PI-DeepONet | $0.58 \pm 0.29$ | $0.46 \pm 0.22$ | $0.37 \pm 0.20$ | $0.36 \pm 0.20$ | $0.32 \pm 0.18$ |
| | | SepONet | $0.62 \pm 0.28$ | $0.35 \pm 0.22$ | $0.32 \pm 0.23$ | $0.28 \pm 0.20$ | $0.29 \pm 0.21$ |
| | Memory (GB) | PI-DeepONet | 0.729 | 1.227 | 3.023 | 9.175 | 35.371 |
| | | SepONet | **0.715** | **0.717** | **0.715** | **0.717** | **0.719** |
| | Training time (hours) | PI-DeepONet | 0.0433 | 0.0641 | 0.1497 | 0.7252 | 2.8025 |
| | | SepONet | **0.0403** | **0.0418** | **0.0430** | **0.0427** | **0.0326** |
| Advection $d=2$ | Relative $\ell_2$ error (%) | PI-DeepONet | $9.27 \pm 1.94$ | $7.55 \pm 1.86$ | $6.79 \pm 1.84$ | $6.69 \pm 1.95$ | $5.72 \pm 1.57$ |
| | | SepONet | $14.29 \pm 2.65$ | $11.96 \pm 2.17$ | $6.14 \pm 1.58$ | $5.80 \pm 1.57$ | $4.99 \pm 1.40$ |
| | RMSE ($\times 10^{-2}$) | PI-DeepONet | $5.88 \pm 1.34$ | $4.79 \pm 1.27$ | $4.31 \pm 1.23$ | $4.24 \pm 1.29$ | $3.63 \pm 1.05$ |
| | | SepONet | $9.06 \pm 1.88$ | $7.58 \pm 1.55$ | $3.90 \pm 1.09$ | $3.69 \pm 1.07$ | $3.17 \pm 0.95$ |
| | Memory (GB) | PI-DeepONet | 0.967 | 1.741 | 5.103 | 17.995 | 59.806 |
| | | SepONet | **0.713** | **0.715** | **0.715** | **0.715** | **0.719** |
| | Training time (hours) | PI-DeepONet | **0.0787** | 0.1411 | 0.4836 | 2.3987 | 8.231 |
| | | SepONet | 0.0843 | **0.0815** | **0.0844** | **0.0726** | **0.0730** |
| Burgers' $d=2$ | Relative $\ell_2$ error (%) | PI-DeepONet | $29.33 \pm 3.85$ | $20.31 \pm 4.31$ | $14.17 \pm 5.25$ | $13.72 \pm 5.59$ | - |
| | | SepONet | $29.42 \pm 3.79$ | $31.53 \pm 3.44$ | $28.74 \pm 4.11$ | $11.85 \pm 4.06$ | $7.51 \pm 4.04$ |
| | RMSE ($\times 10^{-2}$) | PI-DeepONet | $4.19 \pm 2.79$ | $2.82 \pm 1.86$ | $2.23 \pm 2.10$ | $2.20 \pm 2.13$ | - |
| | | SepONet | $4.18 \pm 2.74$ | $4.44 \pm 2.81$ | $4.11 \pm 2.76$ | $1.80 \pm 1.60$ | $1.23 \pm 1.44$ |
| | Memory (GB) | PI-DeepONet | 1.253 | 2.781 | 5.087 | 18.001 | - |
| | | SepONet | **0.603** | **0.605** | **0.603** | **0.605** | **0.716** |
| | Training time (hours) | PI-DeepONet | 0.1497 | 0.2375 | 0.6431 | 3.2162 | - |
| | | SepONet | **0.0706** | **0.0719** | **0.0716** | **0.0718** | **0.0605** |
| Nonlinear diffusion $d=3$ | Relative $\ell_2$ error (%) | PI-DeepONet | $17.38 \pm 5.56$ | $9.90 \pm 2.91$ | - | - | - |
| | | SepONet | $16.10 \pm 4.46$ | $12.11 \pm 3.89$ | $6.81 \pm 1.98$ | $6.73 \pm 1.96$ | $6.44 \pm 1.69$ |
| | RMSE ($\times 10^{-2}$) | PI-DeepONet | $1.86 \pm 0.62$ | $1.04 \pm 0.23$ | - | - | - |
| | | SepONet | $1.72 \pm 0.49$ | $1.29 \pm 0.37$ | $0.72 \pm 0.19$ | $0.71 \pm 0.17$ | $0.68 \pm 0.15$ |
| | Memory (GB) | PI-DeepONet | 6.993 | 37.715 | - | - | - |
| | | SepONet | **3.471** | **2.897** | **2.899** | **2.897** | **13.139** |
| | Training time (hours) | PI-DeepONet | 0.5836 | 6.6399 | - | - | - |
| | | SepONet | **0.1044** | **0.1069** | **0.1056** | **0.1456** | **0.5575** |

Table 7: Performance comparison of PI-DeepONet and SepONet with varying number of input training functions ($N_f$) and fixed number of training points ($N_c = 128^d$, $d$ indicated by problem instance). The '-' symbol indicates that results are not available due to out-of-memory issues.

| Equations | Metrics | Models | 5 | 10 | 20 | 50 | 100 |
|---|---|---|---|---|---|---|---|
| Diffusion-Reaction $d = 2$ | Relative $\ell_2$ error (%) | PI-DeepONet | $34.54 \pm 27.83$ | $4.23 \pm 2.52$ | $1.72 \pm 1.00$ | $0.91 \pm 0.46$ | $0.73 \pm 0.34$ |
| | | SepONet | $22.40 \pm 12.30$ | $3.11 \pm 1.89$ | $1.19 \pm 0.74$ | $0.73 \pm 0.32$ | $0.62 \pm 0.26$ |
| | RMSE ($\times 10^{-2}$) | PI-DeepONet | $14.50 \pm 9.04$ | $1.75 \pm 0.90$ | $0.71 \pm 0.37$ | $0.40 \pm 0.28$ | $0.32 \pm 0.18$ |
| | | SepONet | $9.34 \pm 5.37$ | $1.36 \pm 1.07$ | $0.50 \pm 0.29$ | $0.34 \pm 0.25$ | $0.29 \pm 0.21$ |
| | Memory (GB) | PI-DeepONet | 2.767 | 5.105 | 9.239 | 17.951 | 35.371 |
| | | SepONet | **0.719** | **0.719** | **0.717** | **0.717** | **0.719** |
| | Training time (hours) | PI-DeepONet | 0.1268 | 0.2218 | 0.5864 | 1.4018 | 2.8025 |
| | | SepONet | **0.0375** | **0.0390** | **0.0370** | **0.0317** | **0.0326** |
| Advection $d = 2$ | Relative $\ell_2$ error (%) | PI-DeepONet | $9.64 \pm 2.91$ | $8.77 \pm 2.23$ | $7.57 \pm 1.98$ | $6.69 \pm 1.93$ | $5.72 \pm 1.57$ |
| | | SepONet | $7.62 \pm 2.06$ | $6.59 \pm 1.71$ | $5.47 \pm 1.57$ | $5.18 \pm 1.51$ | $4.99 \pm 1.40$ |
| | RMSE ($\times 10^{-2}$) | PI-DeepONet | $6.11 \pm 1.90$ | $5.55 \pm 1.51$ | $4.80 \pm 1.33$ | $4.24 \pm 1.28$ | $3.63 \pm 1.05$ |
| | | SepONet | $4.83 \pm 1.38$ | $4.18 \pm 1.17$ | $3.47 \pm 1.06$ | $3.29 \pm 1.02$ | $3.17 \pm 0.95$ |
| | Memory (GB) | PI-DeepONet | 3.021 | 5.611 | 9.707 | 34.511 | 59.806 |
| | | SepONet | **0.713** | **0.715** | **0.719** | **0.719** | **0.719** |
| | Training time (hours) | PI-DeepONet | 0.3997 | 1.0766 | 1.9765 | 4.411 | 8.231 |
| | | SepONet | **0.0754** | **0.0715** | **0.0736** | **0.0720** | **0.0730** |
| Burgers' $d = 2$ | Relative $\ell_2$ error (%) | PI-DeepONet | $28.48 \pm 4.17$ | $28.63 \pm 4.10$ | $28.26 \pm 4.38$ | $12.33 \pm 5.14$ | - |
| | | SepONet | $27.79 \pm 4.40$ | $28.16 \pm 4.24$ | $22.78 \pm 6.47$ | $10.25 \pm 4.44$ | $7.51 \pm 4.04$ |
| | RMSE ($\times 10^{-2}$) | PI-DeepONet | $4.09 \pm 2.77$ | $4.11 \pm 2.78$ | $4.07 \pm 2.78$ | $1.96 \pm 1.92$ | - |
| | | SepONet | $4.01 \pm 2.75$ | $4.05 \pm 2.76$ | $3.30 \pm 2.55$ | $1.65 \pm 1.64$ | $1.23 \pm 1.44$ |
| | Memory (GB) | PI-DeepONet | 5.085 | 9.695 | 17.913 | 35.433 | - |
| | | SepONet | **0.605** | **0.607** | **0.607** | **0.609** | **0.716** |
| | Training time (hours) | PI-DeepONet | 0.5135 | 1.3896 | 2.6904 | 5.923 | - |
| | | SepONet | **0.0725** | **0.0707** | **0.0703** | **0.0612** | **0.0605** |
| Nonlinear diffusion $d = 3$ | Relative $\ell_2$ error (%) | PI-DeepONet | - | - | - | - | - |
| | | SepONet | $31.94 \pm 9.18$ | $25.48 \pm 8.95$ | $21.16 \pm 7.82$ | $10.21 \pm 3.31$ | $6.44 \pm 1.69$ |
| | RMSE ($\times 10^{-2}$) | PI-DeepONet | - | - | - | - | - |
| | | SepONet | $3.44 \pm 1.13$ | $2.73 \pm 0.99$ | $2.27 \pm 0.91$ | $1.09 \pm 0.32$ | $0.68 \pm 0.15$ |
| | Memory (GB) | PI-DeepONet | - | - | - | - | - |
| | | SepONet | **2.923** | **3.139** | **4.947** | **6.995** | **13.139** |
| | Training time (hours) | PI-DeepONet | - | - | - | - | - |
| | | SepONet | **0.1175** | **0.1408** | **0.1849** | **0.3262** | **0.5575** |

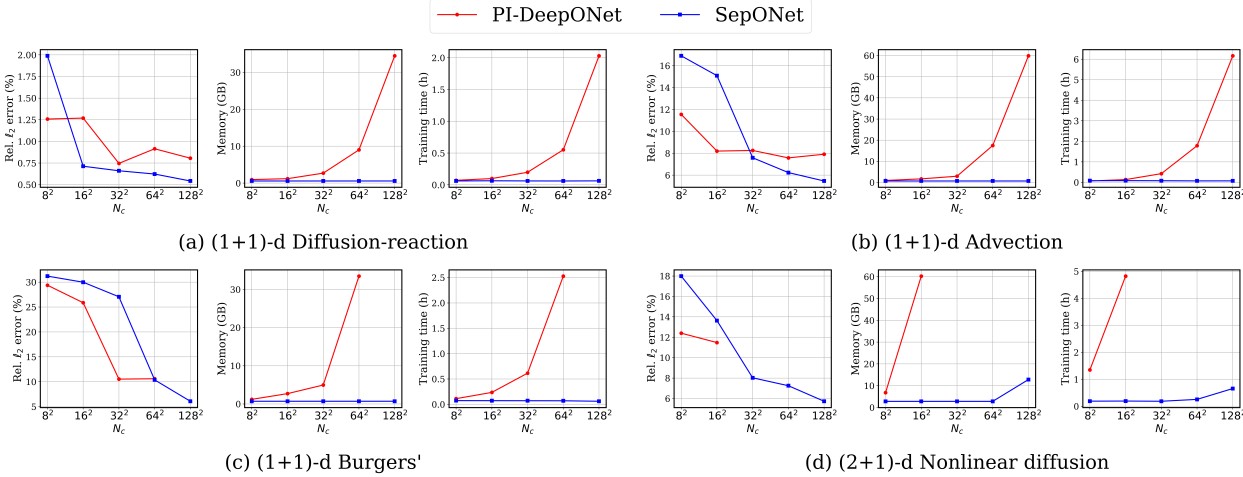

Figure 4: Performance comparison of PI-DeepONet and SepONet with TanH trunk network activation functions, varying number of training points ($N_c$) and fixed number of input functions ($N_f = 100$). Results show test accuracy, GPU memory usage, and training time for four PDEs.

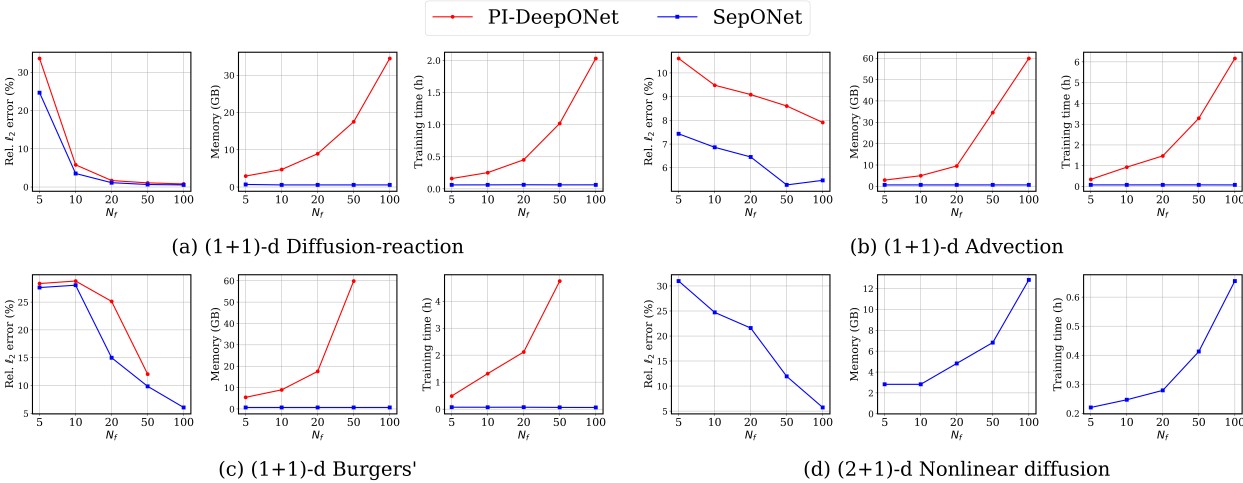

Figure 5: Performance comparison of PI-DeepONet and SepONet with TanH trunk network activation functions, increasing number of input functions ($N_f$) and fixed number of training points ($N_c = 128^d$, where $d$ is the problem dimension). Note: PI-DeepONet results for the (2+1)-dimensional diffusion equation are unavailable due to memory constraints.

## B.4 Ablation Studies

### B.4.1 Trunk Networks with Hyperbolic Tangent Activations

In Figure 4 and Figure 5, we provide complete testing results repeating our experiments from Figure 2 and Figure 3 for all PDE examples, varying $N_c$ and $N_f$, except we use hyperbolic tangenet (TanH) activation functions for all hidden and output layers of the trunk networks in both PI-DeepONet and SepONet. The results are very similar, indicating that alternative activation functions may be chosen for multi-layer trunk networks to maintain the universal approximation property in accordance with Theorem 1.

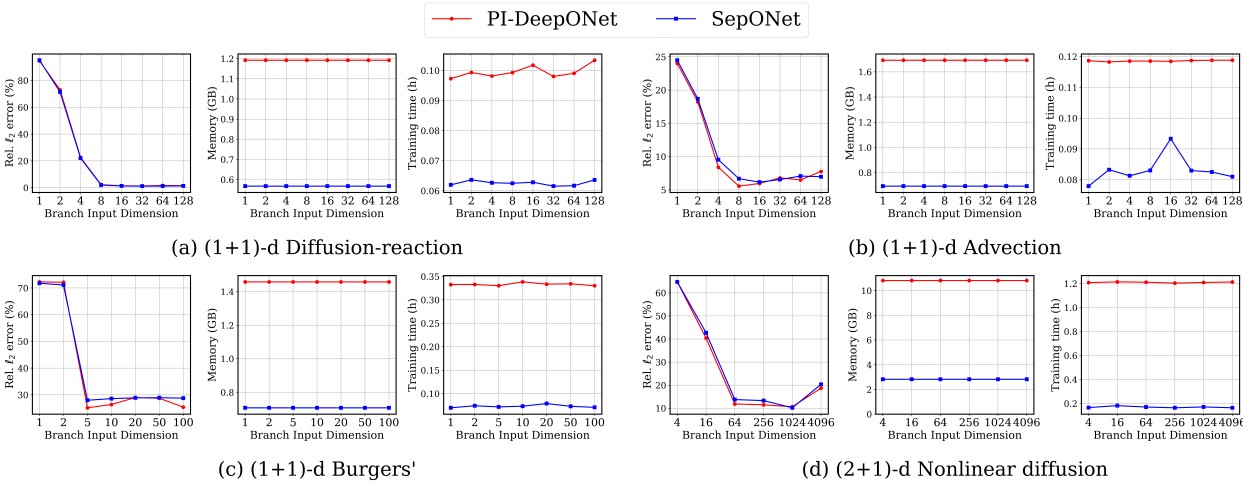

Figure 6: Performance comparison of PI-DeepONet and SepONet with varied number of input function sensors (branch input dimension). Note that we fix $N_f = 20$ for all experiments, $N_c = 32^2$ for (a)-(c), and $N_c = 16^3$ for (d).

### B.4.2 Varying the Input Function Discretization to the Branch Network

Given an input function PDE configuration $u$, recall that the branch network predicts coefficients $\beta_k = b_\psi(\mathcal{E}(u))$ for $k = 1, ..., r$. Our studies in Section 4 use a simple encoder that measures the input function $\mathcal{E}(u) = (u(x_1), ..., u(x_m))$ at points $x_1, x_2, ..., x_m \in K$ in the input function domain. High-dimensional or highly oscillatory input functions may lead to unwieldy discretizations with large branch input dimension that affect training performance. Here, in Figure 6, we study the sensitivity of the PDE examples from Section 4 with respect to the number of input function sensors (branch input dimension). Note that we fix the number of input functions $N_f = 20$ for all experiments, while $N_c = 32^2$ for diffusion-reaction, advection, and Burgers' equations, and $N_c = 16^3$ for nonlinear diffusion. We find that the error curves converge to the minimum value using only a fraction of the number of input sensors that we used in the main text in Figure 2 and Figure 3. This indicates that the input functions we considered may be identified with a small number of points. Nevertheless, we find that training performance in terms of both memory consumption and training time is constant with the number of sensors. This is because training complexity is mainly data-dominated by the need to compute high-order derivatives with respect to a large number of collocation points for evaluation of the physics loss, as discussed in Section 3.2.

## C Complete Solution to Separation of Variables Example (7)

Recall the linear PDE system treated in Section 2.3:

$$\mathcal{M}[t]s(y) = \mathcal{L}_1[y_1]s(y) + \cdots + \mathcal{L}_d[y_d]s(y), \tag{44}$$

where $\mathcal{M}[t] = \frac{d}{dt} + h(t)$ is a first order differential operator of $t$, and $\mathcal{L}_1[y_1], ..., \mathcal{L}_d[y_d]$ are linear second order ordinary differential operators of their respective variables $y_1, ..., y_d$ only. Furthermore, assume we are provided Robin boundary conditions in each variable and separable initial condition $s(t = 0, y_1, \ldots, y_d) = \prod_{n=1}^d \phi_n(y_n)$ for functions $\phi_n(y_n)$ that satisfy the boundary conditions.

Assuming a separable solution exists, $s(y) = T(t)Y_1(y_1) \cdots Y_d(y_d)$, the PDE can be decomposed in the following form:

$$\frac{\mathcal{M}T(t)}{T(t)} = \frac{\mathcal{L}_1 Y_1(y_1)}{Y_1(y_1)} + \cdots + \frac{\mathcal{L}_d Y_d(y_d)}{Y_d(y_d)}, \tag{45}$$

where it is apparent that each term in the sequence is a constant, since they are each only functions of a single variable. Consequently, we may solve each of the $\mathcal{L}_n$ terms independently using Sturm-Liouville

theory. After we have found the associated eigenfunctions $(Y_n^{k_n})$ and eigenvalues $(\lambda_n^{k_n})$, we may manually integrate the left-hand side. Finally, we may decompose the separable initial condition into a product of sums of the orthonormal basis functions (eigenfunctions) of each variable. The resulting solution is given by

$$
\begin{aligned}
s(y) = s(t, y_1, \ldots, y_d) &= \sum_{k=(k_1,\ldots,k_d)} B_k T^k(t) \prod_{n=1}^{d} Y_n^{k_n}(y_n), \\
T^k(t) := T^{(k_1,\ldots,k_d)}(t) &= \exp\left( -\int_0^t h(\tau)d\tau + t \sum_{n=1}^{d} \lambda_n^{k_n} \right), \\
B_k := B_{(k_1,\ldots,k_d)} = \prod_{n=1}^{d} \frac{\langle Y_n^{k_n}(y_n), \phi_n(y_n)\rangle_n}{\langle Y_n^{k_n}(y_n), Y_n^{k_n}(y_n)\rangle_n}&, \quad \lambda_n^{k_n} = \frac{\mathcal{L}_n Y_n^{k_n}(y_n)}{Y_n^{k_n}(y_n)}, \\
n = 1, \ldots, d, \quad k_n &= 1, 2, \ldots, \infty.
\end{aligned}
\tag{46}
$$

Here, $k = (k_1, \ldots, k_d)$, where $k_n \in \{1, 2, \ldots, \infty\}, \forall n \in \{1, \ldots, d\}$, is a lumped index that counts over all possible products of eigenfunctions $Y_n^{k_n}$ with associated eigenvalues $\lambda_n^{k_n}$. $\langle \cdot \rangle_n$ is an appropriate inner product associated with the separated Hilbert space of the $\mathcal{L}_n$-th operator. To obtain equation (7) in the main manuscript, one only need to break up the sum over all $k_n$ indices into a single ordered index.

## D  Additional Experiments

### D.1  (2+1)-dimensional Navier-Stokes Equation

SepONet's memory-efficient and fast-training advantages allow it to tackle more challenging physics-informed operator learning problems, which PI-DeepONet may struggle to train on due to resource constraints. As an example, we consider a (2+1)-dimensional Navier-Stokes equation, previously investigated in the context of PINN (Cho et al., 2024; Wang et al., 2024):

$$
\begin{aligned}
\partial_t \omega + s \cdot \nabla \omega &= 0.01 \Delta \omega, \quad \boldsymbol{x} \in [0, 2\pi]^2, \, t \in \Gamma, \\
\nabla \cdot s &= 0, \quad \boldsymbol{x} \in [0, 2\pi]^2, \, t \in \Gamma, \\
\omega(\boldsymbol{x}, 0) &= \omega_0(\boldsymbol{x}), \, s(\boldsymbol{x}, 0) = s_0(\boldsymbol{x}), \quad \boldsymbol{x} \in [0, 2\pi]^2,
\end{aligned}
\tag{47}
$$

where $s = (s_x, s_y) \in \mathbb{R}^2$ is the velocity field, $\boldsymbol{x} = (x, y)$ denotes 2D spatial variables, $\Gamma = [0, T]$ is the time window, and $\omega = \nabla \times s = \partial_x s_y - \partial_y s_x$ is the vorticity. We aim to learn the solution operator that maps the initial velocity and vorticity field $u(\boldsymbol{x}) = (s_0(\boldsymbol{x}), \omega_0(\boldsymbol{x})) \in \mathbb{R}^3$ to the solution $s(\boldsymbol{x}, t) \in \mathbb{R}^2$ using SepONet, parameterized by $\boldsymbol{\theta} = (\psi, \phi_1, \phi_2, \phi_3)$, which represents all the trainable parameters in the branch and trunk nets. The vector-valued velocity is approximated as:

$$
\begin{aligned}
s_x(u)(\boldsymbol{x}, t) &= \sum_{k=1}^{r} \beta_k \prod_{n=1}^{3} \tau_{n,k}, \\
s_y(u)(\boldsymbol{x}, t) &= \sum_{k=r}^{2r} \beta_k \prod_{n=1}^{3} \tau_{n,k},
\end{aligned}
\tag{48}
$$

where $r$ denotes the rank, $\beta_k = b_\psi(\mathcal{E}(u))_k$ is the $k$-th output of the branch net, and $\tau_{1,k} = t_{\phi_1}^1(t)_k, \tau_{2,k} = t_{\phi_2}^2(x)_k, \tau_{3,k} = t_{\phi_3}^3(y)_k$ denote the $k$-th outputs of the three trunk nets.

**Loss function**  The physics loss for this problem are defined as:

$$
\mathcal{L}_{physics} = \mathcal{L}_{residual} + 5000 \mathcal{L}_{div} + 10000 \mathcal{L}_{initial}.
\tag{49}
$$

The specific loss terms are:

$$
\mathcal{L}_{residual} = \frac{1}{N_f N_r} \sum_{i=1}^{N_f} \sum_{j=1}^{N_r} \left| \frac{\partial (\nabla \times G_{\boldsymbol{\theta}}(u^{(i)})(\boldsymbol{x}_r^{(j)}, t_r^{(j)}))}{\partial t_r^{(j)}} + s^{(i)} \cdot \nabla (\nabla \times G_{\boldsymbol{\theta}}(u^{(i)})(\boldsymbol{x}_r^{(j)}, t_r^{(j)})) \right.
$$

$$
\left. -0.01 \Delta (\nabla \times G_{\boldsymbol{\theta}}(u^{(i)})(\boldsymbol{x}_r^{(j)}, t_r^{(j)})) \right|^2,
$$

$$
\mathcal{L}_{div} = \frac{1}{N_f N_r} \sum_{i=1}^{N_f} \sum_{j=1}^{N_r} \left| \nabla \cdot G_{\boldsymbol{\theta}}(u^{(i)})(\boldsymbol{x}_r^{(j)}, t_r^{(j)}) \right|^2, \tag{50}
$$

$$
\mathcal{L}_{initial} = \frac{1}{N_f N_I} \sum_{i=1}^{N_f} \sum_{j=1}^{N_I} \left( \left| \nabla \times G_{\boldsymbol{\theta}}(u^{(i)})(\boldsymbol{x}_I^{(j)}, 0) - \omega_0^{(i)}(\boldsymbol{x}_I^{(j)}) \right|^2 + \left| G_{\boldsymbol{\theta}}(u^{(i)})(\boldsymbol{x}_I^{(j)}, 0) - s_0^{(i)}(\boldsymbol{x}_I^{(j)}) \right|^2 \right).
$$

Note that periodic boundary conditions are enforced by applying the following positional encoding to the spatial variables:

$$
\gamma(x) = [1, \sin(x), \sin(2x), \sin(3x), \sin(4x), \sin(5x), \cos(x), \cos(2x), \cos(3x), \cos(4x), \cos(5x)]^\top. \tag{51}
$$

**SepONet architecture**   The encoder $\mathcal{E}$ maps the initial condition $u(\boldsymbol{x})$ to its point-wise evaluations at $128 \times 128 \times 3$ sensors on a uniform $128 \times 128$ grid over $[0, 2\pi]^2$. The branch network $b_\psi$ is a CNN, starting with a $1 \times 1$ convolution that increases the channels from 3 to 16, followed by four residual blocks. Each residual block consists of two $3 \times 3$ convolutions with GeLU activations; the first convolution in each block uses a stride of 2 to halve the spatial dimensions while doubling the number of channels. Skip connections employ $1 \times 1$ convolutions with a stride of 2 whenever there is a change in dimension. After flattening, a fully connected layer produces a vector of dimension $2r$.

The trunk networks $t_{\phi_n}$ are 7-layer modified MLPs (Wang et al., 2021a), each with 128 neurons per hidden layer, an input size of 11, and a rank/output size of 256/512, using TanH activations. The initial velocities are sampled from a Gaussian random field with a maximum velocity of 5.

**Training settings**   We consider learning the solution operator within two time windows: $T = 0.1$ and $T = 1$. Two separate SepONet models were trained, each for 100,000 iterations using the Adam optimizer, to minimize the physics loss (49) within their respective time windows. The number of residual points was set to $N_r = 256 \times 256 \times 32$ ($N_x \times N_y \times N_t$), obtained by randomly sampling 256, 256, and 32 points along $x$, $y$, and $t$ axes, respectively, and constructing a mesh grid via the tensor product. The number of initial points, $N_I = 128 \times 128$, corresponds to a uniform mesh grid. Initial velocities were sampled from a Gaussian random field with a maximum velocity of 5. Both the initial conditions and collocation points were resampled every 100 iterations. We varied the number of input functions $N_f$ to see the scaling as data is increased.

**Evaluation**   The model was evaluated on 100 unseen initial conditions, sampled from the same Gaussian random field. Reference solutions for both time windows were obtained using the JAX-CFD solver (Kochkov et al., 2021) on a uniform $128 \times 128 \times 10$ ($N_x \times N_y \times N_t$) grid.

**Results**   The results for two time windows are shown in Figure 7. We find that for $T = 0.1$ we can achieve very low relative $\ell_2$ error of 2.5%. For $T = 1$ the error only scales to 40%. We think that improving the error for the longer time scale represents an interesting application direction for future work. Our architecture and implementation choices were not optimized for this example.

### D.2   Linear Heat Equation Example

SepONet is motivated by the classical method of separation of variables, which is often employed to solve linear partial differential equations (PDEs). To illustrate the connection between these approaches, consider

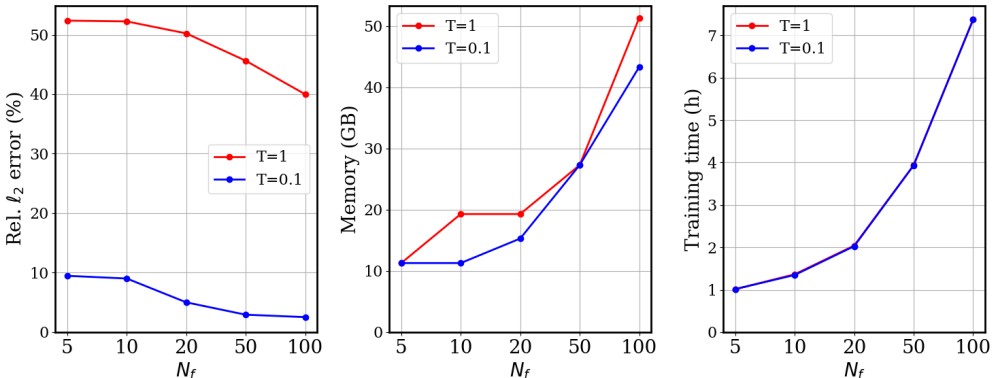

Figure 7: Navier-Stokes equation results with SepONet. Here, we varies the number of input functions $N_f$ and kept the number of collocation points fixed. We consider two cases, $T = 0.1$ and $T = 1$, corresponding to the length of the time window.

the linear heat equation:

$$\begin{aligned}
\frac{\partial s}{\partial t} &= \frac{1}{\pi^2}\frac{\partial^2 s}{\partial x^2}, \quad (x,t) \in (0,1) \times (0,1], \\
s(x,0) &= u(x), \quad x \in (0,1), \\
s(0,t) &= s(1,t) = 0, \quad t \in (0,1).
\end{aligned} \tag{52}$$

The goal is to solve this equation for various initial conditions u(x) using both the separation of variables technique and the SepONet method, allowing for an intuitive comparison between the two.

### D.2.1 Separation of Variables Technique

We seek a solution in the form:

$$s(x,t) = X(x)T(t) \tag{53}$$

for functions $X$, $T$ to be determined. Substituting (53) into (52) yields:

$$\frac{X^{''}}{X} = -\lambda \ \text{ and } \ \frac{\pi^2 T^{'}}{T} = -\lambda \tag{54}$$

for some constant $\lambda$. To satisfy the boundary condition, $X$ must solve the following eigenvalue problem:

$$\begin{aligned}
X^{''}(x) + \lambda X(x) &= 0, \quad x \in (0,1), \\
X(0) = X(1) &= 0,
\end{aligned} \tag{55}$$

and $T$ must solve the ODE problem:

$$T^{'}(t) = -\frac{\lambda}{\pi^2}T(t). \tag{56}$$

The eigenvalue problem (55) has a sequence of solutions:

$$\lambda_k = (k\pi)^2, \quad X_k(x) = \sin(k\pi x), \quad \text{for} \ \ k = 1,2,\dots \tag{57}$$

For any $\lambda$, the ODE solution for $T$ is $T(t) = Ae^{-\frac{\lambda}{\pi^2}t}$ for some constant $A$. Thus, for each eigenfunction $X_k$ with corresponding eigenvalue $\lambda_k$, we have a solution $T_k$ such that the function

$$s_k(x,t) = X_k(x)T_k(t) \tag{58}$$

will be a solution of (54). In fact, an infinite series of the form

$$s(x,t) = \sum_{k=1}^{\infty} X_k(x)T_k(t) = \sum_{k=1}^{\infty} A_k e^{-k^2 t}\sin(k\pi x) \tag{59}$$

will also be a solution satisfying the differential operator and boundary condition of the heat equation (52) subject to appropriate convergence assumptions of this series. Now let $s(x, 0) = u(x)$, we can find coefficients:

$$A_k = 2 \int_0^1 \sin\left(k\pi x\right) u(x) dx \tag{60}$$

such that (59) is the exact solution of the heat equation (52).

### D.2.2 SepONet Method

In this section, we apply the SepONet framework to solve the linear heat equation (52) and compare the basis functions it learns with those derived from the classical separation of variables method. Recall that a SepONet, parameterized by $\boldsymbol{\theta}$, approximates the solution operator of (52) as follows:

$$G_{\boldsymbol{\theta}}(u)(x, t) = \sum_{k=1}^{r} \beta_k(u(x_1), u(x_2), \dots, u(x_{128}))\tau_k(t)\zeta_k(x), \tag{61}$$

where $x_1, x_2, \dots, x_{128}$ are 128 equi-spaced sensors in $[0, 1]$, $\beta_k$ is the $k$-th output of the branch net, and the basis functions $\tau_k(t)$ and $\zeta_k(x)$ are the $k$-th outputs of two independent trunk nets.

**Training settings** The branch and trunk networks each have a width of 5 and a depth of 50. To determine the parameters $\boldsymbol{\theta}$, we trained SepONet for 80,000 iterations, minimizing the physics loss. Specifically, we set $\lambda_I = 20$, $\lambda_b = 1$, $N_f = 100$, and $N_c = 128^2$ in the physics loss. The training functions (initial conditions) $\left\{u^{(i)}\right\}_{i=1}^{N_f}$ were generated from a Gaussian random field (GRF) $\sim \mathcal{N}\left(0, 25^2\left(-\Delta + 5^2 I\right)^{-4}\right)$ using the Chebfun package (Driscoll et al., 2014), ensuring zero Dirichlet boundary conditions. Additional training settings are detailed in Appendix B.2 of the main text.

**Evaluation** We evaluated the model on 100 unseen initial conditions sampled from the same GRF, using the forward Euler method to obtain reference solutions on a $128 \times 128$ uniform spatio-temporal grid.

**Impact of the rank $r$** Since $\tau_k(t)$ and $\zeta_k(x)$ are independent of the initial condition $u$, learning an expressive and rich set of basis functions is crucial for SepONet to generalize to unseen initial conditions. To investigate the impact of the rank $r$ on the generalization error, we trained SepONet with ranks ranging from 1 to 50. The mean RMSE between SepONet's predictions and the reference solutions over 100 unseen test initial conditions was reported. For comparison, we also computed the mean RMSE of the truncated analytical solution at rank $r$ for $r$ from 1 to 15. The results are presented in Figure 8.

As $r$ increases, the truncated analytical solution quickly converges to the reference solution. The nonzero RMSE arises due to numerical errors in computing the coefficients $A_k$ and the inherent inaccuracies of the forward Euler method used to generate the reference solution. For SepONet, we observed that when $r = 1, 2$, the mean RMSE aligns closely with that of the truncated solution. However, as $r$ increases beyond that point, the error decreases more gradually, stabilizing around $r = 50$. This indicates that SepONet may not necessarily learn the exact same basis functions as those from the truncated analytical solution. Instead, a higher rank $r$ allows SepONet to develop its own set of basis functions, achieving similar accuracy to the truncated solution.

**SepONet basis functions** The learned basis functions for different ranks $r$ are visualized in Figure 9 to Figure 13.

At $r = 1$, SepONet learns basis functions that closely resemble the first term of the truncated solution. For $r = 2$, the learned functions are quite similar to the first two terms of the truncated series. However, when $r = 5$, the basis functions diverge from the truncated solution series, although some spatial components still resemble sinusoidal functions and the temporal components remain monotonic. As r increases further, SepONet continues to improve in accuracy, though the learned basis functions increasingly differ from the truncated series, confirming SepONet's ability to accurately approximate the solution using its own learned basis functions.

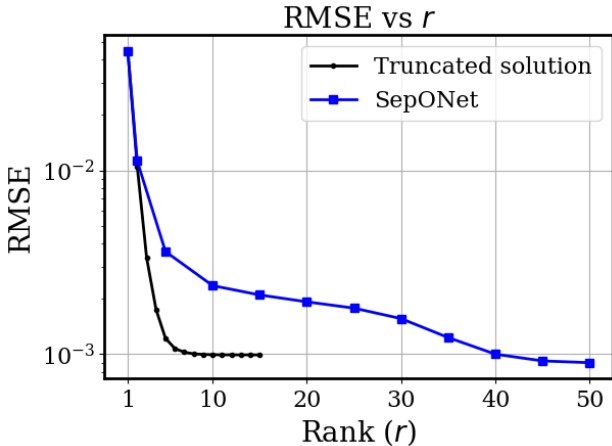

Figure 8: Comparison of RMSE between the truncated analytical solution and SepONet predictions for varying rank $r$. The truncated analytical solution quickly converges, while SepONet shows a slower decay in error, converging around $r = 50$.

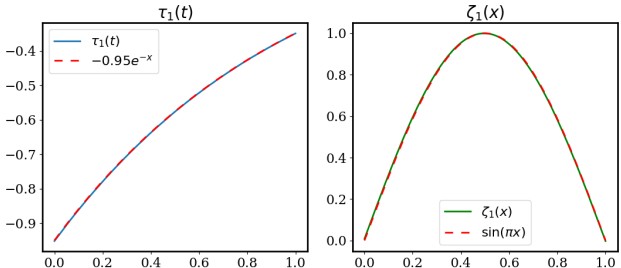

Figure 9: Learned basis functions $\tau_k(t)$ and $\zeta_k(x)$ for $r = 1$. SepONet learns the same basis functions as the first term of the truncated solution.

## E Visualization of SepONet Predictions

In this section, we showcase the performance of trained SepONets in predicting solutions for PDEs under previously unseen configurations. The SepONets were trained using $N_f = 100$ and $N_c = 128^d$, where $d$ denotes the dimensionality of the PDE problem. The prediction results are presented in Figure 14 to Figure 17.

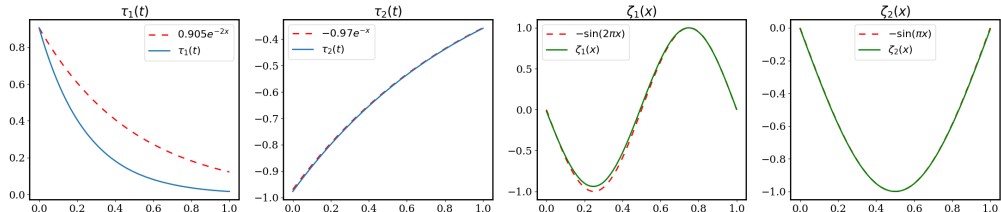

Figure 10: Learned basis functions $\tau_k(t)$ and $\zeta_k(x)$ for $r = 2$. SepONet learns very similar basis functions as the first two terms of the truncated solution.

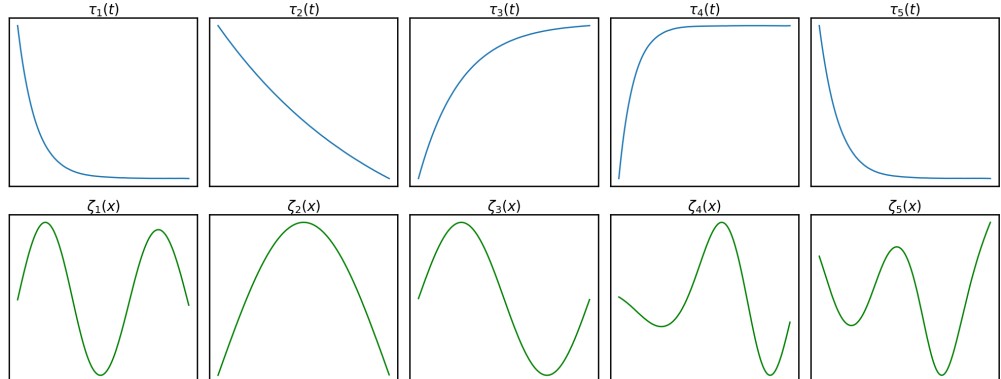

Figure 11: Learned basis functions $\tau_k(t)$ and $\zeta_k(x)$ for $r = 5$.

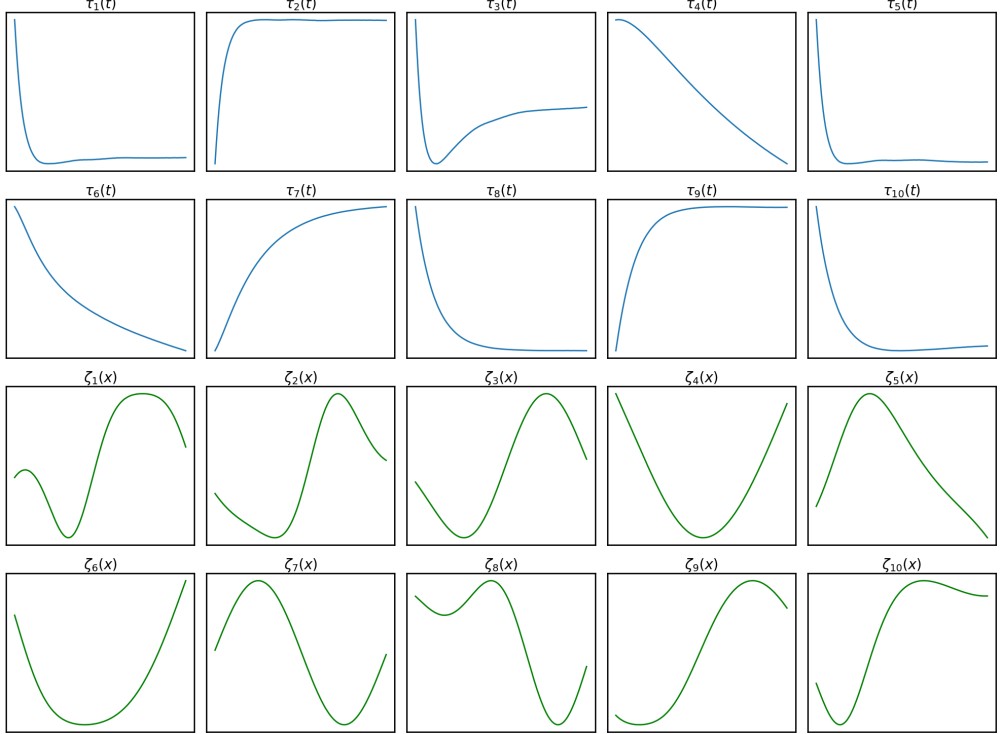

Figure 12: Learned basis functions $\tau_k(t)$ and $\zeta_k(x)$ for $r = 10$.

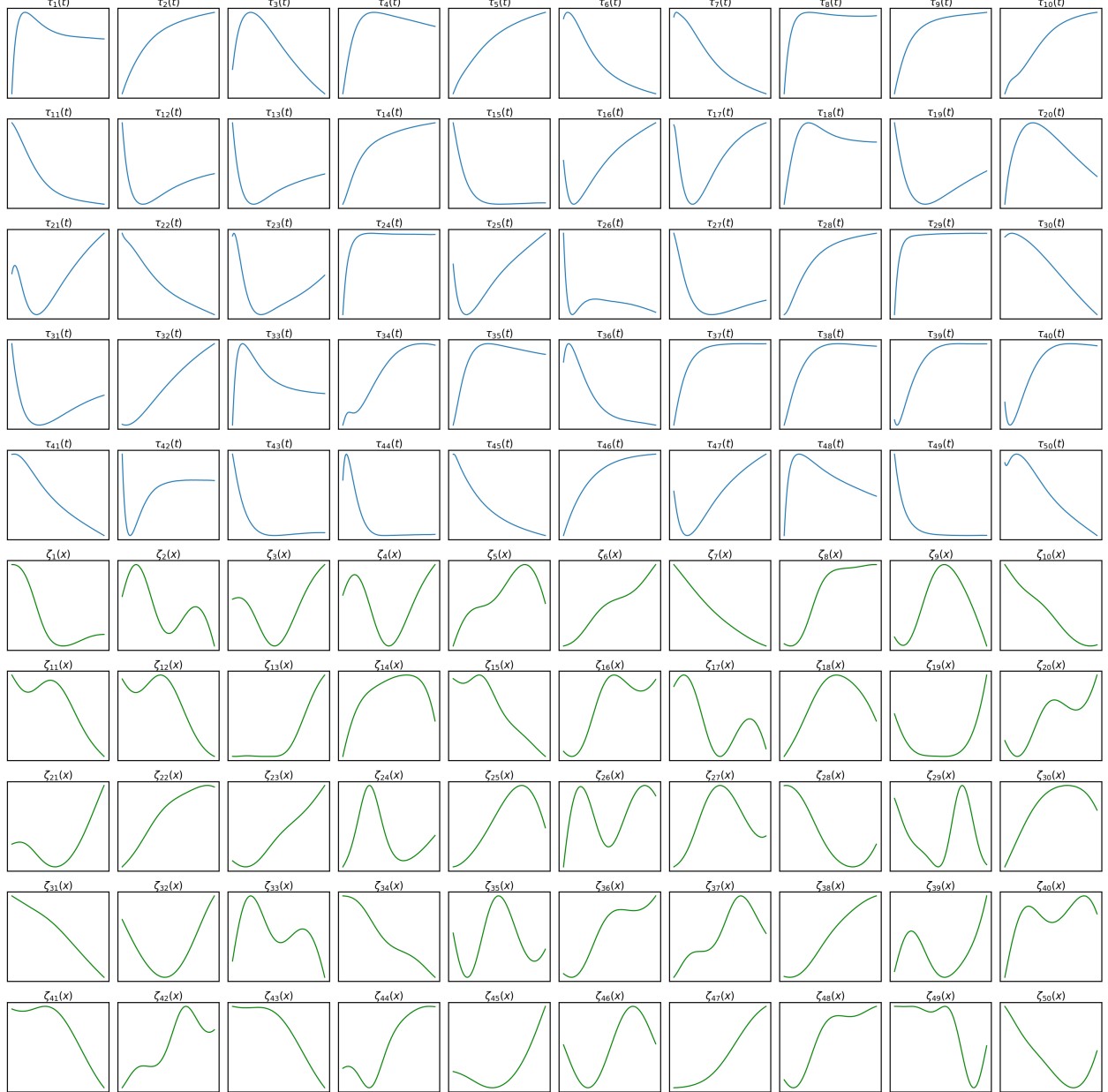

Figure 13: Learned basis functions $\tau_k(t)$ and $\zeta_k(x)$ for $r = 50$.

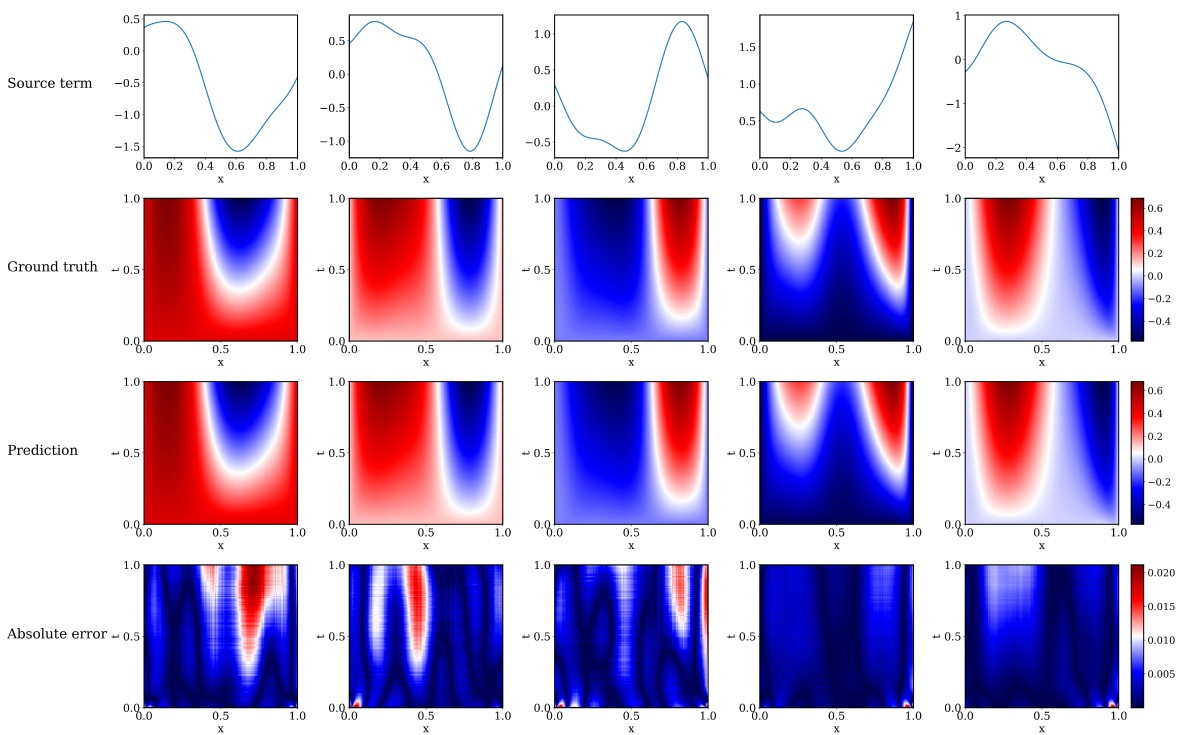

Figure 14: (1+1)-$d$ Diffusion-reaction equation.

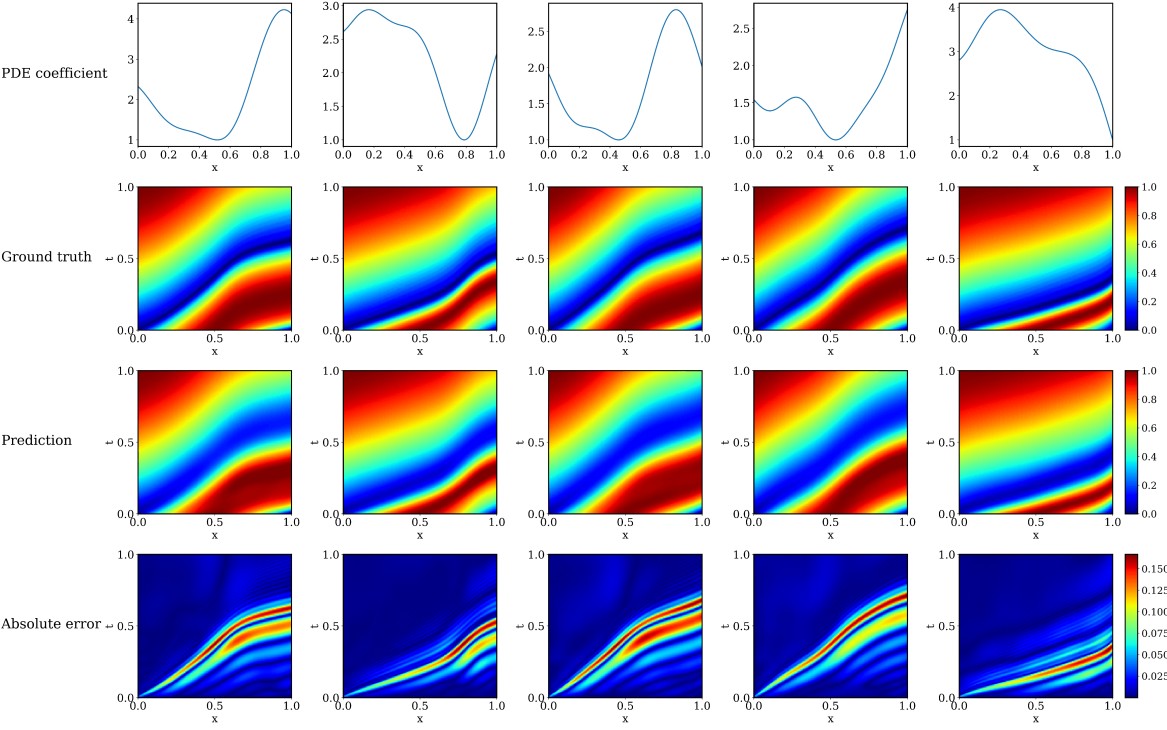

Figure 15: (1+1)-$d$ Advection equation.

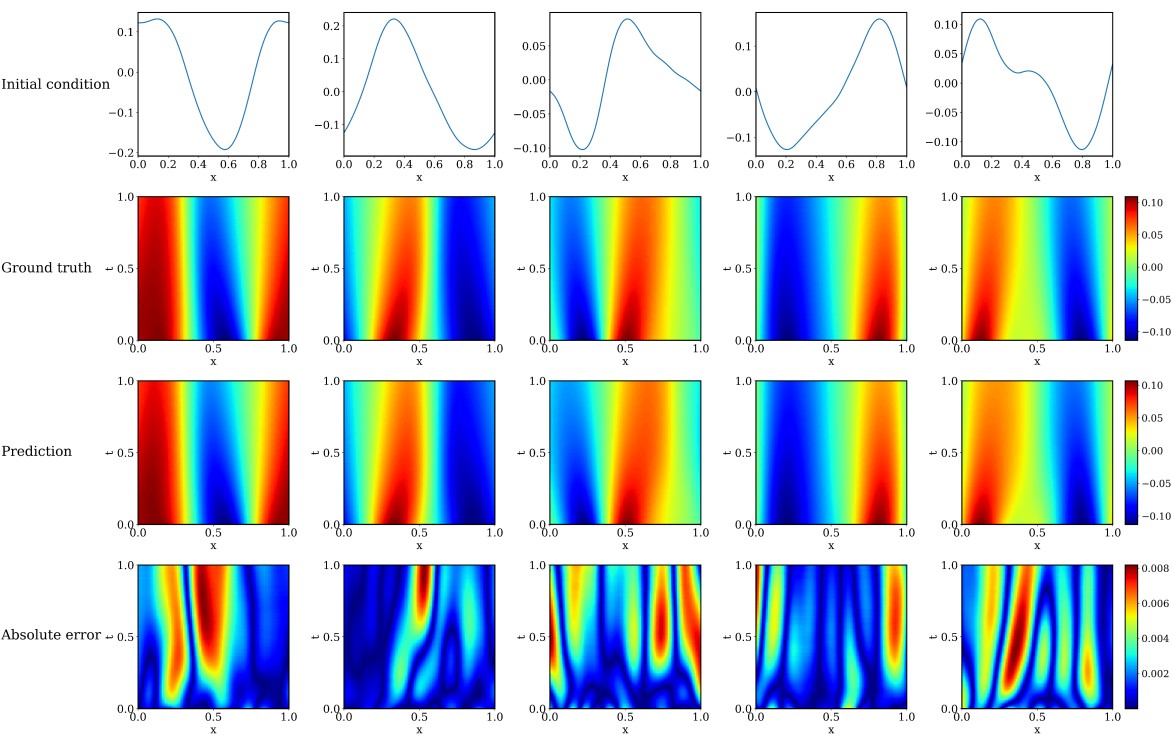

Figure 16: (1+1)-*d* Burgers' equation.

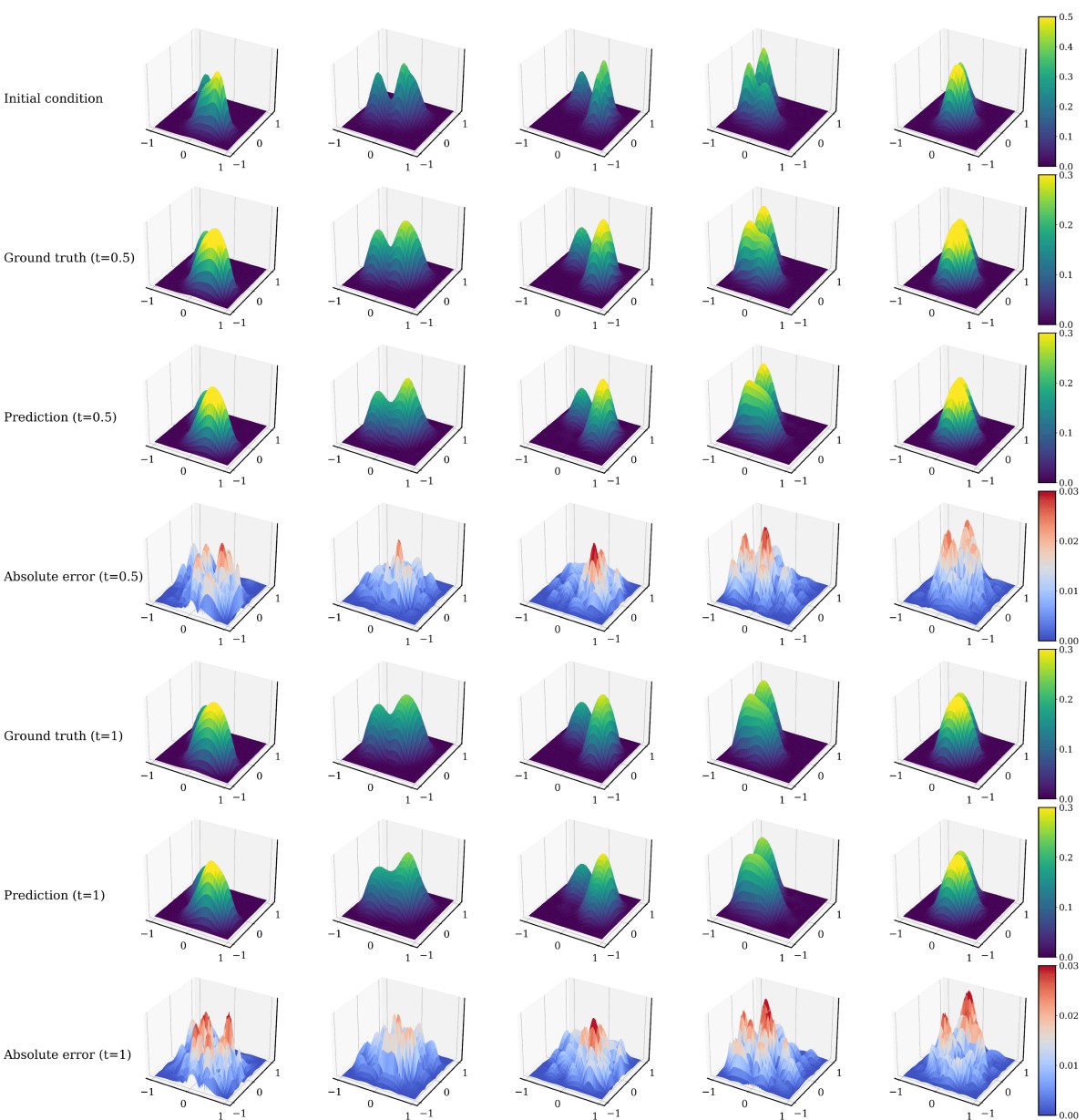

Figure 17: $(2+1)$-$d$ Nonlinear diffusion equation. Two snapshots at $t = 0.5$ and $t = 1$ are presented.

