# OpenReview forum: "Separable Operator Networks"
_TMLR — Accepted by TMLR_

### Review · Reviewer_sLu2 · 2024-10-14

**Summary Of Contributions:**

1. The authors propose a new method called SepONet, which enables efficient physics-informed operator learning by extending the idea of separable PINN to PI-DeepONet.
2. The authors establish the universal approximation capability of the proposed SepONet under proper assumptions.
3. The authors provide an empirical comparison across various PDE problems, showing that SepONet is much more efficient than PI-DeepONet.

**Audience:**

Yes

**Claims And Evidence:**

Yes

**Requested Changes:**

1. Section 3.1.2 needs more clarity about how forward-mode AD helps reduce computation complexity. I agree with the argument about the "input-output relationship" implies that forward-mode AD is more efficient. However, it is unclear from the way it is written right now.
	1. The authors state that computing Eq. (11) with forward-mode AD leads to more efficient computation. However, compute Eq. (11) with reverse-mode AD gives the same time complexity $O(N_m)$.
	2. If I'm not mistaken, the reason why the forward-mode AD is faster in this case is because $\tau_{m}$ is a $\mathbb{R}\rightarrow \mathbb{R}^r$ function and $\partial \tau_m / \partial y_m$  takes one evaluation for forward-mode and $r$ evaluations for reverse-mode.
2. In terms of GPU memory comparison, the authors should provide the full context for precise performance claim.
	1. PI-DeepONet's GPU memory is a linear function of batch size, and it can decrease batch size to reduce GPU memory cost. What is the batch size for PI-DeepONet in the experiments?
	2. Does SepONet use all the collocation points on the grid in one single batch?
3. The authors compare runtime and GPU memory costs of SepONet and PI-DeepONet through concrete experiments, which is good to have. However, as a general operator learning algorithm, the authors should derive the space and time complexities of SepONet in terms of the problem dimensions and compare them with PI-DeepONet, which reveals its efficiency under different scenarios.
4. It would be great to expand the discussion or experiments about vector-valued cases such as Navier-Stokes equations.
5. I think the parenthetical citations should be like (Lu et al., 2022a). However, the authors often write it as Lu et al. (2022a). Just name a few here, but the authors should check the whole paper and correct them.
	1.  Line 8 in the first paragraph of the Introduction: "Lu et al. (2021)"
	2.  Lines 1, 3, and 10 in the third paragraph of the Introduction: "Wang et al. (2021b), Raissi et al. (2019), Baydin et al. (2018)".
6. Minor typos: Eq. (10) needs a comma.

**Strengths And Weaknesses:**

**Strengths**
1. SepONet offers a simple yet effective approach to significantly improve PI-DeepONet's efficiency.

**Weaknesses**
1. Weakness in theoretical results: the proposed theorem assumes the trunk net is one layer with a sinusoidal activation function that does not align with the practical implementation.
2. The efficiency of SepONet heavily relies on the assumption that the grid is regular. If it is an irregular grid, SepONet will no longer have its advantage.
3. The paper does not compare the time and space complexities of SepONet and PI-DeepONet as a function of problem dimensions, which will help us understand the behavior of SepONet under different scenarios as a general algorithm.
4. While the SepONet applies to arbitrary problem dimensions theoretically, the authors only consider the case where the output function is scalar-valued. Common vector-valued cases, such as the Navier-Stokes equation[1], are not studied or discussed.

[1] Jin, Xiaowei, et al. "NSFnets (Navier-Stokes flow nets): Physics-informed neural networks for the incompressible Navier-Stokes equations." _Journal of Computational Physics_ 426 (2021): 109951.

---

> ### Author Response · Authors · 2024-10-31
> **Response regarding weakness 1**
>
> # W1
> We prove that a **one-layer** univariate trunk net with sinusoidal activation function, when combined with a branch net, acts as a universal approximator for nonlinear continuous operators. While the sinusoidal activation is crucial for this one-layer trunk net setup, our theoretical results can be extended to **multi-layer** trunk nets by leveraging the universal approximation theorem to approximate sinusoidal functions. In the previous manuscript submission, we empirically demonstrated that multi-layer trunk nets with sinuosidal activations achieve accurate operator approximations (for both SepONet and PI-DeepONet). In the updated manuscript we have conducted additional experiments with multi-layer trunk networks using Tanh activation functions, which achieves similar accuracy and complexity to the existing cases and supports practical implementations. A subset of our results for the Diffusion-Reaction equation and Advection equation are reproduced in the table below. They are very similar to the results for sinusoidal activation functions in Table 6 of the manuscript.
>
> **Table**: Performance comparison of PI-DeepONet and SepONet with varying number of input training functions ($N_f$), fixed number of training points samples per iteration ($N_c = 128^2$), and TanH trunk net activation functions.
>
> | Equations                        | Metrics                       | Models       | 5             | 10            | 20            | 50            | 100           |
> |----------------------------------|-------------------------------|--------------|---------------|---------------|---------------|---------------|---------------|
> | **Diffusion-Reaction** $d=2$   | Relative $\ell_{2}$ error (%) | PI-DeepONet  | 33.62 ± 28.23 | 5.80 ± 3.60   | 1.68 ± 0.95   | 1.08 ± 0.58   | 0.81 ± 0.42   |
> |                                  |                               | SepONet      | 24.67 ± 17.58 | 3.53 ± 2.26   | 1.13 ± 0.78   | 0.65 ± 0.29   | 0.54 ± 0.23   |
> |                                  | RMSE ($\times 10^{-2}$)     | PI-DeepONet  | 13.69 ± 8.52  | 2.35 ± 1.32   | 0.69 ± 0.33   | 0.47 ± 0.33   | 0.35 ± 0.22   |
> |                                  |                               | SepONet      | 10.13 ± 5.76  | 1.55 ± 1.13   | 0.50 ± 0.37   | 0.30 ± 0.23   | 0.26 ± 0.19   |
> |                                  | Memory (GB)                   | PI-DeepONet  | 2.942         | 4.694         | 8.942         | 17.506        | 34.507        |
> |                                  |                               | **SepONet**  | **0.692**     | **0.567**     | **0.567**     | **0.567**     | **0.567**     |
> |                                  | Training time (hours)         | PI-DeepONet  | 0.1609        | 0.2533        | 0.4528        | 1.0197        | 2.0309        |
> |                                  |                               | **SepONet**  | **0.0617**    | **0.0625**    | **0.0641**    | **0.0625**    | **0.0629**    |
> | **Advection** $d=2$            | Relative $\ell_{2}$ error (%) | PI-DeepONet  | 10.61 ± 3.11  | 9.48 ± 2.15   | 9.09 ± 2.20   | 8.61 ± 2.11   | 7.91 ± 2.01   |
> |                                  |                               | SepONet      | 7.43 ± 2.04   | 6.86 ± 1.77   | 6.45 ± 1.74   | 5.27 ± 1.54   | 5.47 ± 1.55   |
> |                                  | RMSE ($\times 10^{-2}$)     | PI-DeepONet  | 6.72 ± 2.04   | 6.00 ± 1.43   | 5.76 ± 1.46   | 5.45 ± 1.42   | 5.01 ± 1.34   |
> |                                  |                               | SepONet      | 6.11 ± 1.90   | 5.55 ± 1.51   | 4.80 ± 1.33   | 4.24 ± 1.28   | 3.63 ± 1.05   |
> |                                  | Memory (GB)                   | PI-DeepONet  | 2.942         | 5.005         | 9.567         | 34.567        | 59.884        |
> |                                  |                               | **SepONet**  | **0.692**     | **0.692**     | **0.692**     | **0.692**     | **0.692**     |
> |                                  | Training time (hours)         | PI-DeepONet  | 0.3383        | 0.9281        | 1.4709        | 3.2832        | 6.1653        |
> |                                  |                               | **SepONet**  | **0.0738**    | **0.0735**    | **0.0749**    | **0.0746**    | **0.0726**    |
>
> **New supporting data is provided in Figure 4 and Figure 5 of the appendix of the updated manuscript repeating all of our previous experiments with trunk nets that use TanH activation functions along all layers. We have added Remark 4 under Theorem 1 describing how UAT can be used to extend single-layer trunk nets with sinusoidal activation to multi-layer trunk nets with alternative activation.**

---

> ### Author Response · Authors · 2024-10-31
> **response regarding weakness 2 and weakness 3**
>
> # W2
> We acknowledge that SepONet’s current computational advantage is limited to regular grids. To address irregular grids, one option is to divide the irregular grid into subdomains, each approximated by a regular grid (e.g., splitting an L-shaped domain into rectangles). Another option is to apply a coordinate transformation to map the irregular domain onto a regular one (e.g., converting Cartesian to polar coordinates for a circular domain). We will explore these approaches in future work. We'd like to mention that for our experiments provided in Sec. 4, every iteration we sampled points for PI-DeepONet on an irregular grid (uniform random sampling), while points for SepONet in the output function space were sampled on a regular grid (each iteration we sampled a new regular grid of points). We did not observe a notable difference in accuracy between the operators despite the different sampling strategies for these examples.
>
> **We have added clarification through the revised paper, specifically to our implementation in Sec. 3, that the current version of SepONet only has an advantage on regular grids. We have added additional context to the irregular grid implementation in the discussion (Sec. 5).**
>
> # W3
> Thank you for pointing out this important aspect. We have conducted a complexity analysis of the limiting computation of SepONet and PI-DeepONet, specifically when evaluating a physics residual loss, which involves computing at least first-order derivatives of the operator outputs with respect to each corresponding collocation point input to the trunk net.
>
> For SepONet, the collocation points are obtained by randomly sampling $N_n$ points along the $n$-th axis and constructing a Cartesian product grid. For PI-DeepONet, the collocation points are sampled randomly from the entire $d$-dimensional domain, with a total of $M$ points.
>
> Assuming that the output of SepONet has shape $ N_1 \times N_2 \times \dots \times N_d $ and the output of PI-DeepONet has shape $M \times 1$, we analyze the complexity of computing first-order derivatives for all outputs, taking into account the problem dimension $d $, rank $r$, and the number of layers $L$. For simplicity, we assume all trunk nets are $ L$-layer fully connected networks with hidden and output dimensions $ r $, and that $ N_1 = N_2 = \dots = N_d = N $, and $M=N^{d}$. We also compare SepONet using both forward-mode and reverse-mode automatic differentiation (AD). The time and space complexities are summarized in the table below.
>
> **Table**: Time and space complexities of first-order derivatives of SepONet and PI-DeepONet with respect to $ N^d $ residual points using forward-mode and reverse-mode AD.
>
> | **Method**                   | **Time Complexity**                                          | **Space Complexity**                         |
> |------------------------------|--------------------------------------------------------------|----------------------------------------------|
> | SepONet (Forward AD)         | $ O(N \cdot d \cdot L r^2 + N^d \cdot r d) $              | $ O(N \cdot r d + N^d) $                   |
> | SepONet (Reverse AD)         | $ O(N \cdot d \cdot L r^3 + N^d \cdot r d) $              | $ O(N \cdot L r d + N^d) $                 |
> | PI-DeepONet (Reverse AD)     | $ O(N^d \cdot L r^2) $                                    | $ O(N^d \cdot L r) $                       |

---

> ### Author Response · Authors · 2024-10-31
> **response regarding weakness 3 (complexity explanation)**
>
> **Explanation:**
>
> **SepONet with Forward-Mode AD:**
>
> - **Time Complexity:** Derivative computations in the trunk nets take $O(N \cdot d \cdot L r^2)$ time because forward-mode AD computes all $r$ output derivatives in one pass per input. Combining derivatives across $N^d$ grid points via tensor product adds $O(N^d \cdot r d)$.
>
> - **Space Complexity:** Requires $O(N \cdot r d)$ to store trunk net outputs. Forward-mode AD does not store intermediate activations, and combined outputs need $O(N^d)$ space.
>
> **SepONet with Reverse-Mode AD:**
>
> - **Time Complexity:** Reverse-mode AD requires $r$ backward passes per trunk net to compute derivatives for each output dimension, leading to $O(N \cdot d \cdot L r^3)$ time. Combining derivatives via tensor product remains $O(N^d \cdot r d)$.
>
> - **Space Complexity:** Higher memory usage $O(N \cdot L r d)$ due to storing activations during reverse-mode AD.
>
> **PI-DeepONet with Reverse-Mode AD:**
>
> - **Time Complexity:** Processing $N^d$ inputs with input dimension $d$ leads to $O(N^d \cdot L r^2)$.
>
> - **Space Complexity:** Requires $O(N^d \cdot L r)$ to store activations.
>
> From this analysis, in the limiting case of $N^{d-1}\gg Lr$, we observe that both SepONet and PI-DeepONet have time and space complexities that include $N^d$ due to evaluations over all grid points in a $d$-dimensional space. However, the coefficients accompanying the $N^d$ term differ between the two methods. In **SepONet**, the coefficients are $ \cdot r d$ for time complexity and 1 for space complexity. In contrast, in **PI-DeepONet**, the coefficients are $Lr^{2}$ for time complexity and $Lr$ for space complexity. Typically, $d \ll L r$, and therefore, although both methods have exponential scaling with $N^d$, **SepONet is more efficient in practice** due to smaller coefficients in the $N^d$ term. Note that the tensor product in SepONet can be greatly accelerated by GPU parallelization, which is not taken into account in this analysis.
>
> Furthermore, we note that above we have only considered first-order derivatives. Higher-order derivatives are typically computed with similar complexity, since they amount to sequentially repeating the Jacobian-vector products (JVP) (i.e., forward propagating the derivatives of the separated trunk networks multiple times). Indeed, we generally observe a linear time scaling and constant space scaling with the order of the derivatives relative to the complexity analysis above. Furthermore, we did not consider the parameter update in our complexity analysis. In practice, the parameter update (backpropagation of total physics loss) is not limiting for either SepONet or PI-DeepONet.
>
> Finally, note that here we have considered the limiting case with $N^{d-1}\gg Lr$, where the $N^d$ term dominates, we will consider the other case in **RC1** below.
>
> **We have included this complexity comparison in the revised paper** (Sec. 3.2) to provide a clearer understanding of SepONet’s and PI-DeepONet's computational behavior under different scenarios.

---

> ### Author Response · Authors · 2024-10-31
> **response regarding weakness 4**
>
> # W4
> We agree with the reviewer that experiments involving vector-valued output functions would enhance the quality of the manuscript. Please see our results on the 3D Navier-Stokes equation in the table below with vector-valued outputs of dimension 2. $T=0.1$ or $T=1$ correspond to the length of the time window we train and test the operator results from. We also vary the number of input functions $N_f$ to see the scaling as data is increased. We find that for $T=0.1$ we can achieve very low relative $\ell_2$ error of 2.5\%. For $T=1$ the error only scales to 40\%. We think that improving the error for the longer time scale represents an interesting application direction for future work. Our architecture and implementation choices were not optimized for this example.
>
> **Table**: Results for 3D Navier-Stokes equation using SepONet for time windows $T=0.1$ and $T=1$ as we vary the number of input functions $N_f$. In both cases, we sample $256 \times 256 \times 32$ points per iteration in $x$, $y$, and $t$ dimensions.
>
> | Problem Instance        | Metrics                         | 5            | 10           | 20           | 50           | 100          |
> |-------------------------|---------------------------------|--------------|--------------|--------------|--------------|--------------|
> | SepONet ($T=0.1$)     | Relative $\ell_{2}$ error (%) | 9.47 ± 1.92  | 9.02 ± 1.92  | 5.02 ± 1.07  | 2.94 ± 0.61  | 2.54 ± 0.50  |
> |                         | Memory (GB)                     | 11.286       | 11.286       | 15.286       | 27.286       | 43.286       |
> |                         | Training time (hours)           | 1.0161       | 1.3488       | 2.0269       | 3.9335       | 7.3684       |
> | SepONet ($T=1$)       | Relative $\ell_{2}$ error (%) | 52.37 ± 7.06 | 52.24 ± 7.18 | 50.22 ± 7.36 | 45.62 ± 7.33 | 39.99 ± 7.53 |
> |                         | Memory (GB)                     | 11.286       | 19.286       | 19.286       | 27.286       | 51.286       |
> |                         | Training time (hours)           | 1.0126       | 1.3651       | 2.0412       | 3.9372       | 7.3771       |
>
> **We have provided a new experiment studying the Naiver-Stokes equation in Appendix D.1. The data above is plotted in Figure 7 in Appendix D.1.**

---

> ### Author Response · Authors · 2024-10-31
> **response to requested changes**
>
> # RC1
> The time complexities of SepONet (for both forward-mode and reverse-mode AD) and PI-DeepONet are summarized in Table 1. Since $d \ll Lr$ in practical implementation, SepONet demonstrates higher efficiency compared to PI-DeepONet. Regarding forward-mode AD, we agree with the reviewer’s observation: forward-mode AD is faster in SepONet because it computes all $r$ components of the derivatives of the feature function in a single pass, whereas reverse-mode AD requires $r$ separate backward passes. Additionally, forward-mode AD is more memory efficient, as it does not need to store intermediate activations, further contributing to its efficiency in practice.
>
> We note that the SepONet reverse-mode AD analysis assumes that one has optimally implemented the vector-Jacobian product (VJP) by caching the intermediate trunk net outputs. The most straightforward (and naive) implementation of the VJP (e.g., as implemented in JAX) would take the product of a vector of shape $N^d$ with a Jacobian of the full mesh output of the model with shape $N^d\times N$. Hence, one can only obtain $N$ correct derivatives at a time, requiring $N^{d-1}$ VJPs to obtain the derivatives of the full output. Our original discussion of the Jacobian-vector product (JVP) in the manuscript was meant to contrast with this naive case, but we now recognize it was flawed.
>
> **We have provided our complexity analysis in Sec. 3.2. Furthermore, we have improved our notation and discussion of Eq. 10 and Eq. 11 to make it more clear how the JVP is implemented.**
>
> # RC2
> 1. The batch size for PI-DeepONet is $N_f \cdot M$, where $N_f$ is the number of input functions and $M$ is the number of total collocation points. In our experiments, we set $M = N^d$ and randomly sample $M$ collocation points in the computation domain each iteration. We vary both $N_f$ and $N$ to compare efficiency and accuracy. While reducing the batch size lowers GPU memory usage, it also results in a decrease in accuracy.
>
> 2. The batch size for SepONet is $N_f + dN$, but due to the structure of the operator, it can be evaluated at $N_f \cdot N^d$ points in the output function space. The collocation points are obtained by randomly sampling $N$ points along each axis each iteration and constructing the corresponding meshgrid via the tensor product, as described in Eq. (9) from the manuscript.
>
> 3. As indicated in points (1) and (2), both SepONet and PI-DeepONet are evaluated along the same number of points in the output function space, ensuring a fair comparison of GPU memory and runtime performance.
>
> **We have clarified these points in the revised version of the paper to provide full context for our data sampling settings and performance claims.**
>
> # RC3
> Thank you for your suggestion regarding the space and time complexity comparison between SepONet and PI-DeepONet. We have addressed the complexity request above in **W3** and summarized the complexity analysis in the corresponding table. We'd like to make the additional note here that although SepONet and PI-DeepONet scale with $N^d$ in the worst case, in situations where $Lr\gg N^{d-1}$, the first term in SepOnet's time complexity dominates: $O(N\cdot d\cdot Lr^2)$, or in other words, it only scales linearly with dimension and sub-linearly with the total number of collocation points. This situation is not uncommon in many practical 2D and 3D PDEs, considering that commonly $d=2$, $L>3$, $r>50$, and $N<100$ in our experiments.
>
> # RC4
> Thank you for your suggestion. We have expanded our work to include a 3D Navier-Stokes equations example, which can be found in the additional results section in the Appendix D.1. In this example, we address vector-valued cases, specifically involving 2D output functions (velocities). Additionally, we have expanded the discussion on vector-valued cases in the main text to further highlight the applicability of SepONet to such problems.
>
> # RC5
> Corrected, thank you.
>
> # RC6
> Corrected, thank you.

---

### Review · Reviewer_yRAN · 2024-10-16

**Summary Of Contributions:**

The paper proposes a new architecture for operator learning that uses multiple single-dimensional trunk nets to learn and approximate the solution basis function. This proposed model reduces the computational cost associated with high-dimensional grid and brings performance improvement. The proposed method can be useful for physics-informed learning and operator learning tasks. It can also be potentially applicable to some other neural field applications.

**Audience:**

Yes

**Broader Impact Concerns:**

No broader impact concerns.

**Claims And Evidence:**

Yes

**Requested Changes:**

* In SPINN, the authors conduct experiments on arguably more complex PDE problems such as 2+1d Navier-Stokes and multiple 3D systems. Can Sep-DeepONet handle these systems reasonably well? I think it is not a problem even if Sep-DeepONet can't perform well on all of them but it is worth providing some justification and discussion.

* Is the number of collocation point to branch net crucial to the convergence?

* There are some other relevant operator learning works which decompose/factorize the computation leveraging the tensor structure of the data that were not discussed in the paper:

[1] Tran, Alasdair, et al. "Factorized Fourier Neural Operators." The Eleventh International Conference on Learning Representations.

[2] Li, Z., Shu, D., & Barati Farimani, A. (2024). Scalable transformer for pde surrogate modeling. Advances in Neural Information Processing Systems, 36.

[3] Kossaifi, J., Kovachki, N., Azizzadenesheli, K., & Anandkumar, A. (2023). Multi-grid tensorized Fourier neural operator for high-resolution PDEs. arXiv preprint arXiv:2310.00120.

**Strengths And Weaknesses:**

Strengths:

* Represent the the basis of the target function space as tensor product of multiple single-dimensional bases is technically sound. Combined with the forward-mode AD, it allows efficient computation of PDE residuals in physics-informed operator learning.

* The authors extend the previous theoretical result on DeepONet to the case with multiple trunk nets assuming sinusoidal activation in the trunk net.

* Experimental results demonstrate that the proposed Separable DeepONet is much more efficient and can often outperform DeepONet baseline in terms of accuracy.

Weakness:

* The separable learnable bases addresses the curse of dimensionality in the trunk net part but the branch net will still need to handle high-dimensional inputs.

* Separable basis and forward-mode AD are proposed in SPINN, so for this part, the technical advancement is moderate.

---

> ### Author Response · Authors · 2024-10-31
> **response regarding weaknesses and requested changes**
>
> # W1
> We appreciate the reviewer’s comment regarding the potential limitation in the branch network’s handling of high-dimensional inputs. We'd like to emphasize that **this limitation is not unique to SepONet; PI-DeepONet must also address high-dimensional input functions to the branch network**.
>
> It is true that the branch network takes the discretized input functions as its input, which can result in a large input layer, especially when using fine discretizations for high-dimensional input functions. However, since we do not calculate derivatives with respect to the branch net inputs (as they are not needed to evaluate any physics residual losses), the main bottleneck in the current branch net implementation stems from a potentially large input size in the forward pass. To address this, we propose two potential solutions:
>
> 1. **For 1D input functions:** We can reduce the dimensionality of the input by using **coarser discretizations**, which will help in reducing the input size without significantly affecting the model’s performance.
>
> 2. **For higher-dimensional input functions:** A more efficient approach involves using **convolutional neural networks (CNNs)** to process the input functions. CNNs can capture the spatial structure of the input functions while significantly reducing the number of parameters compared to fully connected layers. As demonstrated in recent work such as \cite{mei2024fully}, CNNs provide an efficient way to handle high-dimensional inputs.
>
> To further investigate this, we have conducted new experiments in our paper, showing the performance of SepONet and PI-DeepONet with different discretizations of the input functions for each PDE example (i.e., we have adopted approach (1) above). Interestingly, we find that often only a few features are needed to identify the input function and achieve good accuracy. We'd also like to point out that time and space complexity are not strongly affected by the discretization of the input function.
>
> Additionally, we have implemented a 3D Navier-Stokes PDE example with 3D input functions (2D initial velocities and corresponding vorticity) using a CNN with residual convolutional blocks to process the inputs. More details can be found in the Appendix D.1.
>
> **We have added Figure 6 in Appendix B.4.2 showing the effect of input function discretization size (number of input function sensors).**
>
> # W2
> We appreciate the reviewer’s observation regarding the use of separable bases and forward-mode automatic differentiation (AD), which were introduced in SPINN. We acknowledge that SPINN was an important work that applied these techniques. However, our contribution builds upon SPINN in the following key ways:
>
> 1. **Theoretical Insight and Generalization:** We connect separable basis to the classical method of separation of variables (see Section 2.3 and Appendices C and D), providing deeper intuition. In our framework, individual trunk networks represent basis functions for each variable, and the branch network outputs coefficients for different input functions. This connection allows us to generalize separable basis to a broader class of operator learning problems, not limited to fixed PDE configurations as in SPINN.
>
> 2. **Advancement in Operator Learning:** Unlike SPINN, which requires retraining for each new PDE configuration, our work extends to learning mappings between function spaces—operator learning—enabling the model to generalize across different PDEs without retraining. We also prove a universal approximation theorem for SepONet, demonstrating its ability to approximate nonlinear continuous operators.
>
> # RC1
> We agree that complex systems such as 3D Navier-Stokes equations represents a particularly interesting challenge for operator learning. We have included such an example in Appendix D.1.
>
> # RC2
> Thank you for your question regarding the impact of the number of sensors in the branch network on convergence. Please see our reply in **W1** above. Additionally, here is how we expect the discretization of the input functions to affect performance:
>
> 1. **Low-Frequency, Smooth Input Functions:** Convergence is generally not sensitive to the number of sensors. Even with a smaller number of collocation points, the model can capture the essential features and converge effectively.
>
> 2. **High-Frequency, Oscillatory Input Functions:** The number of sensors becomes crucial. Adequate sampling is necessary to capture complex variations. Insufficient sensors may hinder the model’s ability to accurately identify the input function, affecting convergence.
>
> As mentioned above, we have provided new experiments and analyses showing the performance scaling as we change the discretization of the input function in Figure 6 of the updated manuscript.
>
> # RC3
> Thank you for pointing us to these important references. We have added a discussion of these papers in the introduction of the updated manuscript.

---

> > ### Comment · Reviewer_yRAN · 2024-11-18
> > **Reply to the authors**
> >
> > Thank you for the changes and the detailed response. Most of my concerns have been addressed.

---

### Review · Reviewer_nRRF · 2024-10-20

**Summary Of Contributions:**

This paper tries to motivate a new modification to DeepONets i.e to use a separate trunk net for each variable in the domain of the solution space. The authors give experimental evidence of the efficacy of this method on some small PDEs and they show that this architecture satisfies the same universal approximation theorem as the standard DeepONet.

**Audience:**

Yes

**Claims And Evidence:**

No

**Requested Changes:**

I suggest the following edits,

- In the first paragraph the authors seem to cite (Chen \& Chen 1995) for the universal approximation theorem for operator nets. This is a highly incomplete referencing as the any-depth operator net approximation theorem happened in works like https://doi.org/10.1093/imatrm/tnac001  and https://www.jmlr.org/papers/volume22/21-0806/21-0806.pdf

- At the bottom of page one, the authors say that "to achieve satisfactory generalization error, the number of required input-output function pairs grows exponentially". This statement makes little sense. Generalization bounds in terms of Rademacher complexity already exist for neural operators https://arxiv.org/abs/2205.11359 - where the bounds don't scale exponentially with size at any fixed depth.

    In light of the above reference, the discussion presented needs to be corrected.

 - I think we need much stronger evidence in terms of performance studies on large variable systems - lets say even on the Heat PDE at say dimensions upto 10 or more, lets say as much is the maximum capacity on the free version of Colab-Pro. This is quite needed given that the theoretical contribution is not really novel (this proof is essentially known) and somewhat not connected to the main paper for reasons as explained above.

 - The authors need to *either* (1) show a proof that there exists a net which provably arbitrarily lowers the unsupervised operator net loss as given in equations 4 and 5 or (2) they need to add experiments as described above and relegate the current proof to the Appendix as it is orthogonal to the setup in the rest of the main paper or (3) add experiments in the supervised setup with large variable PDEs which would then correspond to the proof that is given.

 - Also, the way the experiments are presented need to be a lot more clearer. There needs to be a list of PDEs (not just names!) written down which are being tested on. And that needs to be accompanied with the corresponding loss functions and the nets being trained on.

   This clear specification as requested above is really critical to be able to judge the issue of memory footprint of training - which seems to be the main claim here.

**Strengths And Weaknesses:**

There definitely is some merit in the idea of modifying a DeepONet to use a separate trunk net for each variable of the PDE. It is particularly impressive that the performance of this modification has been shown for the unsupervised setting, which is the hard regime for operator learning. But prima-facie this approach endangers needing a huge blow-up in the number of parameters andits not becoming clear from the text as to why this is (probably?) not happening in the implementations.

There are a bunch of unclear things,

- In page 2 there is a count given of $N^d$ collocation points being required. Where is this coming from? It seems to suggest that the only way to use a SepONet is if one gets the collocation points in the domain of the PDE sampled by taking $N$ points in every dimension and then taking the Cartesian product. But its not clear wherefrom in the theory does this constraint on sampling the points comes from. Wont the mechanism work if we randomly sample some points in the domain space of the PDE solutions?

- The experiments are not very convincing given that (A) the PDEs are not very large variable systems and (B) advantage in terms of error is barely there and (C) the claim about lowering of memory footprint is hard to judge because of lack of information on the model sizes in the different experiments.

- The theoretical approximation guarantee that has been presented is somewhat out of place as the proof corresponds to the supervised setting while the experiments and the description in Section 2.2 is in the unsupervised setting.

---

> ### Author Response · Authors · 2024-10-31
> **response to unclear things 1 and 2**
>
> # Unclear thing 1
> Thank you for your question. When optimizing the physics loss at $M$ collocation points, PI-DeepONet always requires $M$ inputs to evaluate PDE solutions and their derivatives, regardless of whether the points are sampled on a regular grid or randomly. In the unsupervised learning setting, we found that using a larger $M$ leads to more accurate and robust solution predictions, as shown in our results. **However, PI-DeepONet becomes inefficient when $M$ is large.**
>
> In contrast, if $M$ can be factorized as $ N_1 \times N_2 \times \ldots \times N_d $, where  $N_n$ is the number of points sampled along the  $n$-th coordinate axis (e.g., $2048 = 8 \times 16 \times 16$ for $d$=3), SepONet only requires $N_1 + N_2 + \dots + N_d $ inputs to evaluate all $M$ collocation points on a regular grid. This factorization allows SepONet to achieve a more efficient solution, both in terms of computation and memory usage.
>
> Additionally, we found that using this per-axis grid-based sampling strategy does not compromise SepONet’s accuracy when compared to PI-DeepONet’s Monte Carlo random sampling across the entire domain.
>
> **In summary, if a large number of randomly sampled $M$ collocation points are needed in PI-DeepONet to achieve reliable predictions, SepONet can employ grid-based sampling for faster and more memory-efficient training, while maintaining the same level of accuracy as PI-DeepONet.**
>
> We have revised all sections related to SepONet’s grid-based sampling requirement to ensure that the motivation for using SepONet is clearly and effectively communicated (specifically in the introduction, Section 1, and implementation, Section 3).
>
> # Unclear thing 2
> (A) While the PDEs we consider are not extremely large variable systems, **solving parametric PDEs in the unsupervised setting is inherently more challenging than in supervised operator learning.** This is because the physics-informed loss must be optimized across multiple PDE configurations. For instance, solving a 2D Advection system took more than 8 hours for PI-DeepONet to reach <6\% relative $\ell_{2}$ error, and PI-DeepONet failed to train the 3D diffusion equation due to memory issues. This is consistent with other works, such as [1], which report long training times. Our goal is to make the first step in scaling up physics-informed DeepONet, and we agree that tackling even larger variable systems is a promising and challenging future direction.
>
> (B) **We do not claim significant accuracy improvements as the main contribution of this paper.** Instead, we focus on showing that SepONet provides substantial reductions in memory and runtime while preserving comparable accuracy to PI-DeepONet. For example, on the 2D advection equation, we provide up to 80x improvement in GPU memory usage and 100x improvement in training time while obtaining similar relative $\ell_{2}$ test error of 5\%. This highlights SepONet’s potential as an efficient and scalable framework for physics-informed operator learning.
>
> (C) We provide the model sizes for different experiments in Appendix B.2, Table 3. Under the same training conditions, SepONet is more memory-efficient than PI-DeepONet, even when it has more parameters due to the use of multiple trunk networks. In all of our experiments, SepONet actually has more parameters than PI-DeepONet (approximately $d$ times more). Indeed, **memory footprint in our experiments is not limited by model size**, but instead by the need to cache data for the computation of partial derivatives along a large number of collocation points in the output function space} (in order to evaluate the physics loss). SepONet relieves this memory bottleneck through the architectural choice of representing the high-dimensional output function as a sum of a tensor-product of univariate basis functions.
>
> **(A) We have added a new 3D example to our experiments (Navier-Stokes) in Appendix D.1. (C) We have added more explicit references to the Appendix listing model parameters. We have added a new complexity analysis showing the time and space complexity scaling for calculating first-order derivatives of PI-DeepONet and SepONet in Section 3.2.**
>
> [1] Liu, Ziyue, et al. "DeepOHeat: operator learning-based ultra-fast thermal simulation in 3D-IC design." 2023 60th ACM/IEEE Design Automation Conference (DAC). IEEE, 2023.

---

> ### Author Response · Authors · 2024-10-31
> **response to unclear thing 3**
>
> # Unclear thing 3
> To clarify, the universal approximation theorem (UAT) we prove for SepONet in this work, as well as the prior work on UAT for DeepONet [1], does **not assume a supervised setting.** The core result of our proof is that there exists a neural operator consisting of a two-layer branch network and $d$ one-layer univariate trunk networks that can approximate any nonlinear continuous operator. This theoretical guarantee is essential for demonstrating that SepONet is a viable neural operator for approximating PDE solution operators.
>
> The method of obtaining the parameters in the branch and trunk networks—whether through supervised data loss or unsupervised physics-based loss—is beyond the scope of this specific theoretical result. Our focus is on establishing that the architecture is capable of approximating the operator, regardless of how the parameters are learned.
>
> **We have added Remark 5 under Theorem 1 in the updated manuscript, indicating that we do not claim any specific scaling laws in our theoretical results.**
>
> [1] Chen, Tianping, and Hong Chen. "Universal approximation to nonlinear operators by neural networks with arbitrary activation functions and its application to dynamical systems." IEEE transactions on neural networks 6.4 (1995): 911-917.

---

> ### Author Response · Authors · 2024-10-31
> **response to requested changes**
>
> # RC1
> Thank you for pointing out the incomplete referencing of the universal approximation theorem for operator networks. We have updated our references to include the works you provided for a more complete and accurate citation.
>
> # RC2
> The original statement was incorrect. The number of required input-output function pairs grows quadratically, not exponentially, as supported by Rademacher complexity bounds for neural operators. We’ve corrected this in the revision.
>
> # RC3
> Please see our reply above in **Unclear things 2**, where we have included a Navier-Stokes equation example with vectorial output as another example of a large-scale PDE.
>
> We agree that exploring high-dimensional PDE systems is a challenging and worthwhile task. We certainly are interested in pursuing such tasks in future work. However, for operator learning problems, we are not aware of any existing benchmarks in the literature with dimensions that large. Furthermore, the 2D and 3D cases we consider in the paper are already exhaustive for PI-DeepONet.
>
> # RC4
> We thank the reviewer for the suggestions to increase the clarity and impact of this paper. We'd like to reiterate that **Theorem 1 in our paper is one of existence**, and does not attempt to provide explicit error bounds with respect to any loss function nor to provide any prescription of how to scale and update model parameters}. Hence, we believe that our proof is appropriate for the main text and not orthogonal to the setup.
>
> We acknowledge that despite the existence of a separable operator solution for any nonlinear continuous operator, we have not provided a theoretical guarantee that the accuracy of the SepONet solution scales tractably with either number of inputs or number of net parameters, for example when provided a physics loss function. Error bounds for DeepONet with supervised loss functions have been derived previously in, e.g., [1]. However, we are not aware of any work that provides error bounds for physics-informed operator learning, assuming a PI-DeepONet or otherwise. In spite of this, we have provided extensive experimental results proving that SepONet accuracy scales at least as well as PI-DeepONet for many practical PDE examples, and we included examples where SepONet scales even when PI-DeepONet fails due to out-of-memory issues. Thus, the result that SepONet is competitive with PI-DeepONet in terms of accuracy and outperforms PI-DeepONet in terms of computational complexity is significant, given that PI-DeepONet is the standard method used by the field for physics-informed operator learning.
>
> Lastly, we would like to mention that we are mainly interested in unsupervised training using physics loss functions in this work. We have pointed to additional references that consider supervised loss function using tailored architectures in the updated manuscript. We agree that looking into supervised learning with SepONet would be another interesting future direction.
>
> **In the revision, Remark 5, we have clarified that Theorem 1 does not imply any neural scaling law nor error bound for SepONet. We have provided a new complexity analysis for physics-informed learning comparing PI-DeepONet and SepONet in Section 3.2, making our claims about the advantages of SepONet for physics-informed operator learning more visible. We have added additional references to Section 1 describing concurrent work on tailored architectures for supervised operator learning.**
>
> # RC5
> All PDE definitions, net definitions, and training hyperparameters were provided in Appendix B. To enhance clarity, in Section 4 we have added a table summarizing the PDE examples in the main paper. We have also added the explicit loss functions for each PDE in Appendix B. Please also note that we will provide open-sourced code upon acceptance of the paper, with explicit implementations of all experiments. The code is currently not included to maintain anonymity.
>
> [1] Lanthaler, Samuel, Siddhartha Mishra, and George E. Karniadakis. "Error estimates for deeponets: A deep learning framework in infinite dimensions." Transactions of Mathematics and Its Applications 6.1 (2022): tnac001.

---

> ### Comment · Reviewer_nRRF · 2024-10-31
> **Questions (Round 2)**
>
> Thanks a lot for the extensive effort to make things clearer.
> The tables 6 and 7 definitely help a lot to understand exactly what is happening.
>
> -  But what is not becoming clear is if the model size is changing as $N_c$ or $N_f$ is changing for any of the rows.
>
>    Where is this critical information being given? Table 5 does not make it clear!
>
>    The authors in their responses seem to clarify that SepONet uses bigger models than PI-DeepONet.
>
>    But is the model size the same for all $N_c$ and $N_f$ values for each PDE?  Is the claim that for the model size remaining the same (at all $N_c$ or $N_f$), SepONet uses less memory than PI-DeepONet despite its model needed being larger?
>
>    There are too many inter-dependencies that are not getting disentangled.
>
> - If the decrease in memory footprint is coming because of the architectural change then we need to see plots where $(N_c,N_f)$ is held fixed and we have pairs of plots, one for SepONet and one for PI-DeepONet, such that in each plot we can see the memory footprint of each method against changing model sizes - and all models being trained to similar accuracy.
>
>    Because SepONet uses bigger models its curve would exist to the right of the curve for DeepONet - but hopefully, the range of this curve on the y-axis will reveal the surprising fact that its memory usage is still low - what is being claimed. Is it possible to see this without too much rerunning of experiments?
>
> - I think the confusion in the writing of the theorem persists. This theorem is just like any existing UAT for operator nets - and it justifies the usage of the architecture in the supervised setting - where each training data is a function pair of the form $(u,G(u))$. And that is fine! The problem here is that the writing does not make clear that such UATs do not justify the usage of the architecture in the unsupervised setting as is true for the experiments here.
>
>
>   I think the authors need to clearly separate out the two parts of the paper - experiments in the unsupervised setting and theory in the supervised setting while making it clear that one does not have a direct bearing on the other. This logical separation can be done even by just clarifying this in the list of results given in the introduction and the heading of Section 4. Instead what we have now is this confusing line on page 3 "We validate our theoretical results through benchmarking SepONet against..."- which is not a correct summary of what is happening.

---

> ### Author Response · Authors · 2024-10-31
> **response to questions (round 2)**
>
> # Q1
> Thank you for your question. Here, $N_c$ and $N_f$ refer to the number of sampled collocation points and the number of sampled functions during training, respectively. You can think of of $N_f$ like the "mini-batch size" for the branch network. Meanwhile, $N_c$ is related to the number of mini-batched inputs to the trunk network(s). **These values do not affect the model architecture or model size.**
>
> **For each PDE, the model size remains fixed regardless of changes in $N_c$ or $N_f$** (model size parameters are reported in Table 5). Both PI-DeepONet and SepONet have branch and trunk networks of the same size; the primary difference is that SepONet employs $d$ independent trunk networks, one for each axis.
>
> Although SepONet’s model is larger due to these multiple trunk networks, it requires less memory during training than PI-DeepONet, and this memory advantage becomes more pronounced as $N_c$ or $N_f$ increases. One of the major reasons for this is that, for given $N_c$ and $N_f$, the number of inputs to SepONet is much smaller than PI-DeepONet. Furthermore, much more memory is consumed during backpropagation for PI-DeepONet compared to forward-AD for SepONet. The GPU memory that we report in this paper is the total memory of data flow during training, not just the model size.
>
> We will clarify the model size in the results section.
>
> # Q2
> Thank you for your suggestion. **The model architecture remains unchanged as $N_c$ or $N_f$ varies**. As we clarified in response to the previous question, for both SepONet and PI-DeepONet, the model size is fixed across different values of $N_c$ and $N_f$. SepONet’s memory efficiency stems from its separable architecture, which enables it to evaluate the physics loss efficiently, as detailed in the complexity analysis in Section 3.2, allowing it to handle higher values of $N_c$ and $N_f$ without a significant increase in memory footprint.
>
> # Q3
> Thank you for your question. We would like to clarify that the original Universal Approximation Theorem (UAT) for operator learning [1], as well as our UAT for SepONet, **does not assume a supervised learning setting or the presence of training function pairs $\left\(u_{i}, G(u_{I})\right\)_{i}^{N_f}$.** This differs from prior work, such as [2], which establishes a specific generalization error bound under a supervised learning setting. To derive this error bound, their approach [2] assumes particular network sizes and a certain number of supervised function pairs. However, this does not imply that the UAT is limited to supervised learning.
>
> **Our theorem is a theoretical result that demonstrates SepONet’s representational capacity**—specifically, it proves the existence of a SepONet configuration capable of approximating any continuous operator to an arbitrary degree of accuracy, given sufficient network complexity. Whether or not we have access to $G(u)$ in any practical setting is irrelevant to the proof.
>
> The motivation for proving this theorem is to show that such a configuration exists in theory. However, this result does not address the practical task of finding or learning the optimal parameters (weights and biases) for SepONet. It is independent of the training paradigm and holds regardless of whether SepONet is trained in a supervised, unsupervised, or reinforcement learning setting, among others.
>
> We appreciate the suggestion to clarify this distinction in the paper. **We agree that the sentence "We validate our theoretical results through benchmarking SepONet against..." is misleading, since our theoretical results do not imply any training setting.** Our intention of this sentence was to indicate that we provide evidence that SepONet has a high representational capacity on nonlinear continuous operator learning problems. In response, we will revise the introduction and results sections to more clearly separate the theoretical aspects of our UAT from the practical, unsupervised experiments. This revision should improve the logical flow and help prevent potential misunderstandings.
>
> [1] Chen, Tianping, and Hong Chen. "Universal approximation to nonlinear operators by neural networks with arbitrary activation functions and its application to dynamical systems." IEEE transactions on neural networks 6.4 (1995): 911-917.
>
> [2] Lanthaler, Samuel, Siddhartha Mishra, and George E. Karniadakis. "Error estimates for deeponets: A deep learning framework in infinite dimensions." Transactions of Mathematics and Its Applications 6.1 (2022): tnac001.

---

### Author Response · Authors · 2024-10-31
**paper revision**

Dear editor and reviewers,

We are extremely grateful for the high-quality reviews and feedback that we have received for this manuscript. We have provided replies to all points raised by the reviewers with corresponding revisions to the manuscript. We have also included a redlined version of the manuscript in supplementary material so that all changes may be visible. We hope that we have sufficiently addressed all criticisms, and fervently believe that the revisions have greatly improved the impact of the manuscript. A summary of the major changes to the manuscript follow:

1. We have revised all sections related to SepONet’s grid-based sampling requirement to ensure that the motivation for using SepONet is clearly and effectively communicated.

2. We have added additional ablation studies showing the effect of using TanH activation functions on all hidden and output layers of the trunk networks of SepONet, with corresponding comparisons to PI-DeepONet.

3. We have added additional ablation studies showing the effect on accuracy and computational performance when varying the discretization of measurements of the input functions to the branch network for both SepONet and PI-DeepONet.

4. We have added an additional 3D Navier-Stokes equation PDE example with vector-valued output for SepONet.

5. We have added theoretical time and space complexity analyses with corresponding discussion of what regimes and by what factors we expect SepONet to computationally outperform PI-DeepONet.

6. We have added an additional remark to our universal approximation theorem for SepONet that we do not claim to have proved tractable neural scaling laws nor error bounds with respect to parameters and data.

---

> ### Author Response · Authors · 2024-11-03
> **paper revision 2**
>
> Dear editor and reviewers,
>
> We once again thank all reviewers for the exceptional feedback we have received for this manuscript, which has helped us greatly improve its quality and clarity. We have updated the revision, as well as the redlined version in the supplementary materials. The main changes in the latest revision were that we have updated the abstract and our contributions to more carefully delineate our theoretical contributions from our experimental results on pages 1-3, and to provide more clarity to the model size and how we calculate GPU memory in Section 4.
>
> Kind regards,
> The Authors

---

### Decision · Action_Editor_RWLT · 2024-11-28

**Recommendation:** Accept as is

**Comment:**

See above.

**Audience:**

The paper is relevant for both the machine learning and physics community.
It improves on existing works such as PI-DeepONet and will interest this community.

**Claims And Evidence:**

In this paper the authors introduce Separable Operator Networks for physics-informed operator learning.
The approach introduces trunk networks to learn different basis functions on different axes.
Additionally the authors prove some theoretical properties regarding the universal property of the introduced architecture.
All the reviewers and myself agree that the paper is sound and supported with experiments.